# ST4VLA: Spatially Guided Training for Vision-Language-Action Model

**Jinhui Ye**[2,1*], **Fangjing Wang**[3,1*], **Ning Gao**[1*], **Junqiu Yu**[1,4*], **Yangkun Zhu**[1], **Bin Wang**[1]
**Jinyu Zhang**[1,4], **Weiyang Jin**[1], **Yanwei Fu**[4], **Feng Zheng**[3], **Yilun Chen**[1†], **Jiangmiao Pang**[1†]

[1]Shanghai AI Laboratory   [2]The Hong Kong University of Science and Technology
[3]Southern University of Science and Technology   [4]Fudan University

## ABSTRACT

Large vision–language models (VLMs) excel at multimodal understanding but fall short when extended to embodied tasks, where instructions must be transformed into low-level motor actions. We introduce ST4VLA, a dual-system **V**ision–**L**anguage–**A**ction framework that leverages **S**patial Guided **T**raining to align action learning with spatial priors in VLMs. ST4VLA includes two stages: (i) spatial grounding pre-training, which equips the VLM with transferable priors via scalable point, box, and trajectory prediction from both web-scale and robot-specific data, and (ii) spatially guided action post-training, which encourages the model to produce richer spatial priors to guide action generation via spatial prompting. This design preserves spatial grounding during policy learning and promotes consistent optimization across spatial and action objectives. Empirically, ST4VLA achieves substantial improvements over vanilla VLA, with performance increasing from 66.1 to 84.6 on Google Robot and from 54.7 to 73.2 on WidowX Robot, establishing new state-of-the-art results on SimplerEnv. It also demonstrates stronger generalization to unseen objects and paraphrased instructions, as well as robustness to long-horizon perturbations in real-world settings. These results highlight scalable spatially guided training as a promising direction for robust, generalizable robot learning. Source code, data and models are released at `https://internrobotics.github.io/internvla-m1.github.io`.

## 1 INTRODUCTION

Large multimodal foundation models Li et al. (2024b); Chen et al. (2024); Bai et al. (2025b); Ye et al. (2025a); Radford et al. (2021); Zhai et al. (2023); Liu et al. (2025b) have demonstrated remarkable generalization capabilities by learning from web-scale vision–language data. However, a critical gap remains when transferring these capabilities to the physical domain, because robots must not only understand *what* an instruction means but also determine *where* and *how* to act in the 3D world. This gap is fundamental, as real-world robotic tasks must align textual instruction with embodiment-specific motor actions. However, textual instruction is sparse, whereas real-world actions demand continuous, embodied interactions. Yet, such text-to-action pairs are inherently scarce in standard VLM training data.

Core spatial priors, such as object recognition, affordance grounding, visual trajectory reasoning, and relative localization, provide transferable and generalizable knowledge for robotic manipulation. Once these spatial priors are established, embodiment-specific learning can focus on concrete control strategies (e.g., manipulator joints, end-effector trajectories, humanoid locomotion, or mobile navigation). Such a division clarifies the role of spatial priors as general-purpose foundations while leaving embodiment-specific details to downstream adaptation, thereby bridging the gap between abstract linguistic instruction and grounded physical execution.

Prior work has approached this challenge through hierarchical robotic systems Huang et al. (2023; 2024a); Liu et al. (2024); Huang et al. (2024b); Qi et al. (2025); Cao et al. (2025); Yuan et al. (2024),

---

*Equal contribution
†Corresponding author

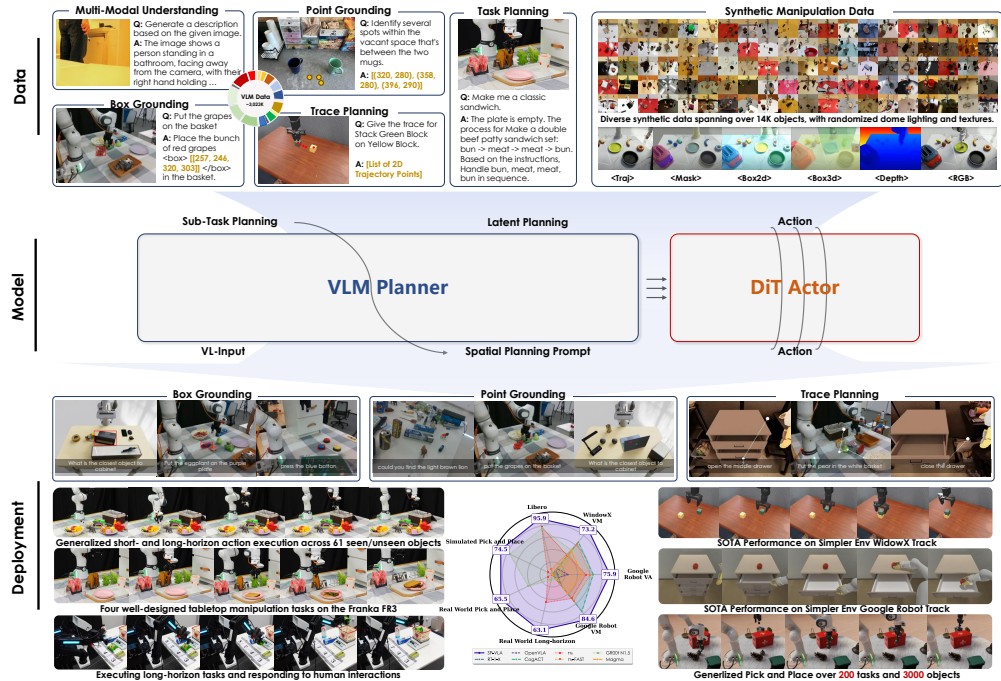

Figure 1: ST4VLA integrates spatial priors into the vision–language–action training pipeline. Given a task instruction, the VLM planner produces latent plans through explicit spatial prompting, which then effectively guides the action expert to generate control signals.

which explicitly encode spatial priors using foundation models Fang et al. (2023); Kirillov et al. (2023); Oquab et al. (2023). However, these methods often rely on rule-based task decomposition and manually designed planning heuristics. The rigid separation between symbolic task structures and low-level motor control makes it difficult to scale automatically to complex and diverse tasks, and particularly limits the potential for end-to-end policy learning. In contrast, recent data-driven VLAs Kim et al. (2024); Brohan et al. (2023); Black et al. (2024); Shi et al. (2025); AI (2024); Lee et al. (2025) leverage pretrained vision-language models and large-scale teleoperation datasets Collaboration et al. (2023); Khazatsky et al. (2024); Bu et al. (2025a); Wu et al. (2024); starVLA Contributors (2025) to directly learn robot control. While these approaches remove the need for manual task heuristics, they tend to overfit low-level motor patterns and thus fail to fully exploit spatial priors during execution. Our empirical analysis in Figure 3 further confirms this limitation: naive fine-tuning VLM to VLA or joint training with spatial data yield weak alignment between spatial perception and action-learning objectives, which undermines spatial grounding during policy learning.

To address the fundamental gap between multimodal understanding and embodied control, we propose **ST4VLA**, a dual-system vision–language–action framework that explicitly integrates *spatial priors* into robot control through spatial guided training. Unlike prior approaches that either rely on rule-based task decomposition or overfit to low-level motor patterns, ST4VLA strategically separates *where and what to act* from *how to act*, ensuring reliable and generalizable manipulation. At its core, ST4VLA introduces a two-stage training pipeline. 1) *spatial grounding pre-training*, the VLM planner acquires transferable spatial priors (point, box, trajectory) by unifying web-scale multimodal grounding data with robot-specific datasets, thereby equipping the model with affordance grounding, localization, and trajectory reasoning. 2) *spatially guided action post-training*, the action expert is conditioned on these spatial priors through lightweight spatial prompting, aligning optimization between perception and control while preserving the VLM's grounding capacity.

This spatially guided training recipe offers three key benefits. First, it preserves spatial grounding during policy learning, thereby avoiding the collapse observed in naive co-training. Second, it aligns the optimization dynamics of multimodal perception and action objectives, resulting in more stable and robust learning. Third, it enhances generalization to unseen objects, novel instructions, and

long-horizon tasks in real-world settings. Empirically, ST4VLA achieves state-of-the-art results on SimplerEnv benchmarks, improves large-scale simulation tasks by over 6% on average, and reaches 92% success on real-world long-horizon manipulation under distribution shifts. These findings highlight spatially guided training as a scalable and reliable paradigm for generalist robot learning.

This work makes the following contributions:

- We observe that directly fine-tuning a VLM with an action expert as a VLA model leads to a collapse of spatial priors, and that naïve co-training with spatial data introduces gradient conflicts between spatial grounding and action objectives. In contrast, simple spatial prompting effectively mitigates these issues (Section 3.1).
- We propose ST4VLA, a spatially guided training framework that explicitly aligns action optimization with spatial grounding objectives, preserving perception while enabling robust control.
- We present a comprehensive evaluation of ST4VLA, establishing leading performance on large-scale simulation and real-robot experiments benchmark. ST4VLA substantially improves generalization to unseen objects, novel instructions, and out-of-distribution environments, outperforming strong baselines such as $\pi_0$ Black et al. (2024) and GR00T Bjorck et al. (2025).

## 2 METHODS

We propose ST4VLA, a spatially guided training framework that bridges spatial understanding with embodied control through a novel two-stage training recipe 2.2. As shown in Figure 2, our approach decouples the acquisition of spatial priors from embodiment-specific control, enabling robust instruction following in diverse and complex scenes.

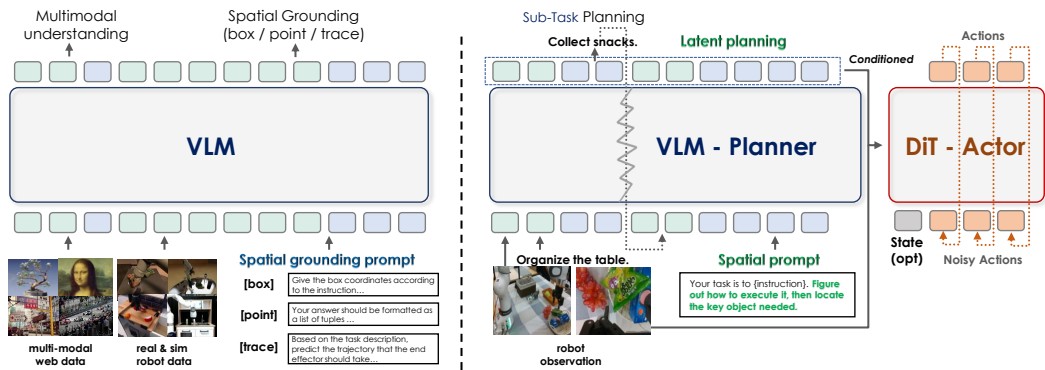

Figure 2: **Overview of ST4VLA.** ST4VLA adopts a spatially guided two-stage training pipeline. Stage 1 (spatial grounding pre-training): the VLM is trained on large-scale multisource multimodal spatial grounding data to learn embodiment-agnostic spatial priors. Stage 2 (spatially guided action post-training): the VLM Planner, functioning as a slow but reliable System 2 reasoner, generates latent planning tokens via spatial prompting as the condition to the action expert (instantiated as a DiT Actor) to execute as a fast System 1 embodiment-specific controller.

### 2.1 MODEL ARCHITECTURE

**Dual-system, Dual-supervision.** We propose a new a dual-system, end-to-end VLA framework based on Qwen2.5-VL, which can foster alignment between the optimization dynamics of the spatial grounding objective and the action policy objective. The framework builds on a dual-system architecture: System 2 (the VLM planner) employs as a multimodal encoder to capture spatial and semantic priors, while System 1 (the Action Expert) adopts a compact diffusion transformer (DiT) Peebles & Xie (2023) and a DINOv2 visual encoder Oquab et al. (2023) for embodiment-specific control.

Specifically, the VLM planner is aligned with a broad range of spatial grounding data, both real and synthetic, covering tasks such as object detection, affordance recognition, and visual trajectory

planning. In parallel, the Action Expert is trained on robot demonstration data, enabling it to specialize these priors into embodiment-specific motor commands. This dual-supervision strategy establishes a cohesive link between high-level semantic perception and low-level motion control, which is essential for robust instruction following in both simulation and real-world settings.

To connect the VLM Planner with the action expert, we adopt a lightweight querying transformer (8.7 MB) conditioned on the latent spatial grounding embeddings produced by the VLM Planner. The querying transformer stabilizes expert learning and inference by mapping variable-length input tokens into a fixed set of learnable query tokens. It is implemented as a $k$-layer cross-attention module, where the query tokens selectively attend to $k$ intermediate layers of the VLM (e.g., $k = 1$ attends only to the final layer).

**Latent grounding via spatial prompting.** To explicitly activate the spatial perception capability learned during spatial grounding pre-training, we employ spatial prompting during post-action training stage. For instance, in general object manipulation tasks, we append simple prompts such as "Figure out how to execute it, then locate the key object needed" after the task instruction. The extracted feature embeddings provide the planner with explicit spatial cues that facilitate more reliable grounding. Motivated by prior studies Driess et al. (2025); Zhou et al. (2025b); Bjorck et al. (2025) showing that direct gradient flow between action and VLM modules may distort multimodal knowledge, we introduce a gradient decay factor within the querying transformer. This attenuates the gradients propagated from the Action Expert back to the VLM (e.g., by a factor of 0.5), thereby preserving the Planner's semantic reasoning ability while still enabling effective joint optimization.

## 2.2 TRAINING RECIPE

To leverage spatial priors for stronger embodiment-specific control in diverse scenarios, ST4VLA adopts a spatially guided two-stage training pipeline:

**Stage 1: Spatial grounding pre-training.** The objective of the first stage (see Figure 2) is to establish a foundational alignment between generic visuo-linguistic understanding and the specific spatial reasoning demands of robotics, thereby priming the model for the subsequent co-adaptation of grounding and action objectives. To this end, we strategically combine large-scale internet vision-language grounding corpora (e.g., RefCOCO Yu et al. (2016), LLaVA-OneVision Li et al. (2024a)) with targeted robot-specific datasets (e.g., RoboRefIt Lu et al. (2023), A0 Xu et al. (2025b), and ST4VLA Data). This combination ensures that the VLM's spatial priors are not only grounded in broad visual concepts but are also directly relevant to robotic tasks such as bounding-box detection, affordance recognition, and trajectory prediction. By reformatting all robotic data into a unified QA structure consistent with web-scale pre-training, we enable the VLM to develop a spatially-aware representation space under a standard supervised fine-tuning framework, which serves as a synergistic foundation for joint optimization with the action policy.

**Stage 2: Spatially guided action post-training.** This stage focuses on learning embodiment-specific control while maintaining and refining the spatial priors acquired in Stage 1. Beyond co-training with spatial grounding data, where the VLM backbone is updated via next-token prediction on image-prompt pairs, we further introduce spatial prompting for action data to enhance alignment between semantic reasoning and motion generation. For action sequences, we augment the standard task instruction with a spatial prompt that elicits the VLM's internal reasoning about scene geometry; for example, the instruction "store all toys into the toy box" is extended to "Identify all relevant toys and their spatial relationships to the container."

## 3 EXPERIMENTS

We conduct comprehensive experiments to evaluate whether aligning the optimization dynamics of multimodal grounding and action policy objectives enables robust robot manipulation. First, we perform a preliminary study to explore the alignment between spatial grounding and action learning during training (Section 3.1). Next, we assess performance on public simulated benchmarks to establish competitive baselines (Section 3.2). We then evaluate large-scale instruction-following pick-and-place in simulation and real-world to test generalization (Section 3.3 and Section 3.4 ). Finally, we examine real-robot performance on both short-horizon and long-horizon tasks to validate practical deployment capabilities (Section 3.5).

## 3.1 PRELIMINARY: PERCEPTION-ACTION CO-OPTIMIZATION

To systematically investigate whether spatial grounding capabilities influence the manipulation performance of VLAs, we track the co-optimization of spatial perception and manipulation success during training. Furthermore, inspired by Raghu et al. (2017); Fang et al. (2024), we quantify the alignment between the two objectives using similarity between gradient matrices. We compare three distinct training strategies using the OXE dataset for action data and a curated set of spatial grounding datasets for multimodal co-training:

- **Vanilla VLA**: direct fine-tuning of a pre-trained VLM on manipulation data only.
- **Vanilla Co-training VLA**: joint optimization on both spatial grounding data and action data.
- **Spatially Guided Training VLA (ST4VLA)**: incorporates spatially pretrained and spatial prompting during pretraining and co-training with multimodal data during post-training.

**Empirical experiment analysis.** Figure 3 (a) and Figure 3 (b) illustrate the interaction between manipulation success (WidowX) and perception performance (on RefCOCO-g) across training steps. Vanilla VLA shows rapid spatial perception degradation, with RefCOCO-g performance dropping to near-random levels by 20k steps, indicating that action-only optimization disrupts spatial representations. Vanilla co-training partially preserves perception but exhibits unstable oscillations in both metrics. Our Spatially Guided approach achieves the best balance: it maintains 70% of original RefCOGO-g performance while reaching 60% WidowX success in just 20k steps.

These trends are further substantiated by the comprehensive benchmark results in Table 1. Compared to the vanilla co-training baseline, our ST4VLA achieves superior robotic manipulation performance (84.6% VM / 75.9% VA on Google Robot and 73.2% on WidowX) while simultaneously preserving stronger multimodal perception and spatial grounding capabilities across all evaluated tasks.

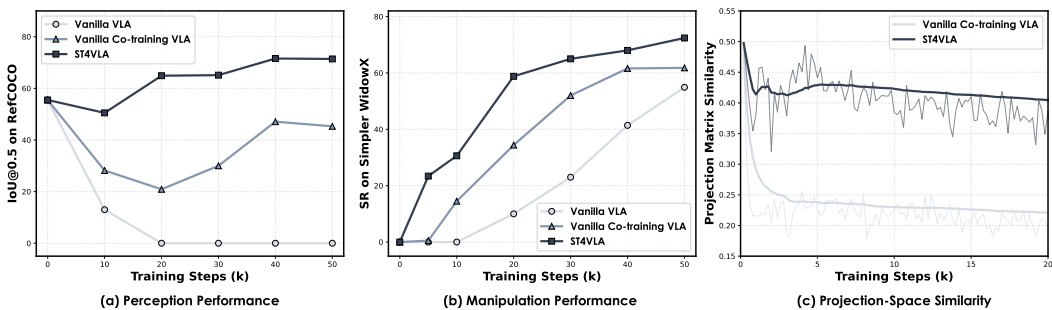

Figure 3: Ablation study on the effect of auxiliary spatial prompting during co-training. From left to right: (a) perception performance (IoU@0.5 on RefCOCO-g); (b) manipulation performance (Average Success Rate on WidowX); (c) shows the gradient similarity of the spatial grounding and action policy objectives, when taking vanilla co-training or the proposed spatially prompting co-training.

Table 1: Study of VLA training strategies and their effects on multi-modal understanding, spatial grounding, and robot manipulation performance.

| Models | Multi-modal Understanding | | | | Spatial Grounding | | | | Robotic Manipulation | |
|---|---|---|---|---|---|---|---|---|---|---|
| | MME | MMVet | TextVQA OCR | POPE Acc | COCO Caption BLEU/ROUGE_l | Refcoco-g IoU0.5 | Where2place point-Acc | Refit-testB Acc0.5 | Google Robot VM/VA | WidowX VM |
| Vanilla VLA | - | - | - | - | - | - | - | - | 66.1/63.5 | 54.7 |
| Vanilla co-train | 1106 | 19.2 | 20.5 | 78.0 | 10.4/15.1 | 47.1 | 21.4 | 66.7 | 70.2/66.5 | 61.1 |
| +Spatially Guided | 1374 | 23.0 | 28.4 | 84.6 | 13.0/13.7 | 68.1 | 25.5 | 72.5 | 78.8/70.0 | 67.4 |
| +Spatially Pretrained | 1411 | 23.3 | 28.6 | 86.2 | 13.0/13.4 | 71.2 | 25.5 | 74.3 | 84.6/75.9 | 73.2 |

**Gradient dynamics analysis.** We introduce Projection-Space Similarity (PSS) Raghu et al. (2017), a method to quantify the alignment between the optimization dynamics of the multimodal grounding objective and the action policy objective. The core idea is to compare the gradients induced by each objective on a shared set of model parameters. Higher PSS values indicate better subspace alignment

between the two optimization processes, validating that action policy optimization coherently builds upon spatial representations. Further methodological details are provided in Appendix Section A.

As shown in Figure 3(c), vanilla co-training of action data with spatial data yields a PSS of only 0.25, indicating significant misalignment between the gradient subspaces. In contrast, our spatially guided training approach increases the PSS to 0.42, demonstrating substantially improved optimization consistency. This enhanced alignment correlates with better preservation of spatial perception capabilities and faster convergence in manipulation tasks.

## 3.2 EXPERIMENTS ON PUBLIC BENCHMARK

We evaluate ST4VLA on the SimplerEnv simulation suite to assess its robustness to visual appearance shifts in instruction-following tasks. SimplerEnv includes both WidowX and Google Robot platforms, short-horizon atomic tasks, and controlled variations in lighting, color, surface texture, and camera pose. We report results on three task sets: Google Robot-VM (visual matching under viewpoint and lighting changes), Google Robot-VA (visual aggregation with varying textures and colors), and WidowX-VM (cross-robot generalization). We further evaluate ST4VLA on the LIBERO simulation suite, detailed in Appendix Section B.2

Table 2: Result comparisons of robotic manipulation on SimplerEnv (Google-Robot) benchmark. The underlined scores indicate the best results excluding ST4VLA. Numbers are officially reported; otherwise, we reimplement and mark such entries with $*$.

| Google Robot | Models | Co-Train | Pick Coke Can | Move Near | Open/Close Drawer | Open Top Drawer and Place Apple | Avg |
|---|---|---|---|---|---|---|---|
| Visual Matching | RT-1 Brohan et al. (2022) | ✗ | 85.7 | 44.2 | 73.0 | 6.5 | 52.4 |
| | RT-1-X Collaboration et al. (2023) | ✗ | 56.7 | 31.7 | 59.7 | 21.3 | 42.4 |
| | RT-2-X Brohan et al. (2023) | ✓ | 78.7 | 77.9 | 25.0 | 3.7 | 46.3 |
| | OpenVLA Kim et al. (2024) | ✗ | 18.0 | 56.3 | 63.0 | 0.0 | 34.3 |
| | CogACT Li et al. (2024c) | ✗ | 91.3 | 85.0 | 71.8 | 50.9 | 74.8 |
| | SpatialVLA Qu et al. (2025) | ✗ | 86.0 | 77.9 | 57.4 | - | 75.1 |
| | $\pi_0$ Black et al. (2024) | ✗ | 72.7 | 65.3 | 38.3 | - | 58.8 |
| | $\pi_0$-FAST Pertsch et al. (2025) | ✗ | 75.3 | 67.5 | 42.9 | - | 61.9 |
| | GR00T N1.5* Bjorck et al. (2025) | ✗ | 51.7 | 54.0 | 27.8 | 7.4 | 35.2 |
| | Magma Yang et al. (2025a) | ✓ | 83.7 | 65.4 | 56.0 | 6.4 | 52.9 |
| | Vanilla VLA | ✗ | 90.0 | 69.8 | 52.5 | 52.2 | 66.1 |
| | Vanilla Co-training VLA | ✓ | 91.3 | 75.1 | 55.0 | 59.4 | 70.2 |
| | **ST4VLA** | ✓ | **97.3** | **98.0** | **65.3** | **77.8** | **84.6** |
| Variant Aggregation | RT-1 Brohan et al. (2022) | ✗ | 89.8 | 50.0 | 32.3 | 2.6 | 43.7 |
| | RT-1-X Collaboration et al. (2023) | ✗ | 49.0 | 32.3 | 29.4 | 10.1 | 30.2 |
| | RT-2-X Brohan et al. (2023) | ✓ | 82.3 | 79.2 | 35.3 | 20.6 | 54.4 |
| | OpenVLA Kim et al. (2024) | ✗ | 60.8 | 67.7 | 28.8 | 0.0 | 39.3 |
| | CogACT Li et al. (2024c) | ✗ | 89.6 | 80.8 | 28.3 | 46.6 | 61.3 |
| | SpatialVLA Qu et al. (2025) | ✗ | 88.0 | 82.5 | 41.8 | - | 70.7 |
| | $\pi_0$ Black et al. (2024) | ✗ | 75.2 | 63.7 | 25.6 | - | 54.8 |
| | $\pi_0$-FAST Pertsch et al. (2025) | ✗ | 77.6 | 68.2 | 31.3 | - | 59.0 |
| | GR00T N1.5 Bjorck et al. (2025) | ✗ | 69.3 | 68.7 | 35.8 | 4.0 | 44.5 |
| | Magma Yang et al. (2025a) | ✓ | 68.8 | 65.7 | 53.4 | 18.5 | 51.6 |
| | Vanilla VLA | ✗ | 92.3 | **80.3** | 50.1 | 31.4 | 63.5 |
| | Vanilla Co-training VLA | ✓ | 82.6 | 73.5 | 62.4 | 47.5 | 66.5 |
| | **ST4VLA** | ✓ | **95.6** | 74.5 | **68.0** | **65.3** | **75.9** |

**Baselines.** We compare to state-of-the-art open VLA systems, including $\pi_0$ Black et al. (2024), GR00T Bjorck et al. (2025), OpenVLA Kim et al. (2024), CogACT Li et al. (2024c), and etc. We also include a Vanilla VLA built on `Qwen2.5-VL-3B-Instruct` with a DiT action expert. When available, we use official reported numbers; otherwise, we reimplement and mark such entries with $*$. We keep training data, observation spaces, and action type aligned with the most popular setups Li et al. (2024c) to ensure a fair comparison.

**Result Analysis.** The main experimental results are presented in Table 2 and Table 3. Compared with prior state-of-the-art models, it attains a 5.9% gain in Google Robot Visual Matching, a 5.3% gain in Visual Aggregation, and a 9.8% gain on the WidowX benchmark. These results highlight the strong competitiveness of ST4VLA within the community. Compared to the Vanilla VLA based on QwenVL-2.5-3B-Instruct, ST4VLA achieves substantial improvements: a 14.6% increase in Google Robot Visual Matching and a 12.4% increase in Visual Aggregation, along with a 17.0%

Table 3: Result comparisons of robotic manipulation on SimplerEnv (WidowX) benchmark. The underlined scores indicate the best results, excluding our results.

| WidowX Robot | Models | Co-Train | Put Spoon on Towel | Put Carrot on Plate | Stack Green Block on Yellow Block | Put Eggplant in Yellow Basket | Avg |
|---|---|---|---|---|---|---|---|
| | RT-1-X Brohan et al. (2022) | ✗ | 0.0 | 4.2 | 0.0 | 0.0 | 1.1 |
| | Octo-Base Octo Model Team et al. (2024) | ✗ | 15.8 | 12.5 | 0.0 | 41.7 | 17.5 |
| | Octo-Small Octo Model Team et al. (2024) | ✗ | 41.7 | 8.2 | 0.0 | 56.7 | 26.7 |
| | OpenVLA Kim et al. (2024) | ✗ | 4.2 | 0.0 | 0.0 | 12.5 | 4.2 |
| Visual | CogACT Li et al. (2024c) | ✗ | 71.7 | 50.8 | 15.0 | 67.5 | 51.3 |
| Matching | SpatialVLA Qu et al. (2025) | ✗ | 16.7 | 25.0 | 29.2 | 100.0 | 42.7 |
| | $\pi_0$ Black et al. (2024) | ✗ | 29.1 | 0.0 | 16.6 | 62.5 | 27.1 |
| | $\pi_0$-FAST Pertsch et al. (2025) | ✗ | 29.1 | 21.9 | 10.8 | 66.6 | 48.3 |
| | GR00T N1.5 Bjorck et al. (2025) | ✗ | 75.3 | 54.3 | 57.0 | 61.3 | 61.9 |
| | Magma Yang et al. (2025a) | ✓ | 37.5 | 31.0 | 12.7 | 60.5 | 35.8 |
| | Vanilla VLA | ✗ | 56.6 | 63.3 | 27.0 | 71.8 | 54.7 |
| | Vanilla Co-trainig VLA | ✓ | 70.3 | 68.4 | 20.5 | 85.2 | 61.1 |
| | **ST4VLA** | ✓ | **80.2** | **79.2** | **35.4** | **98.0** | **73.2** |

improvement on the WidowX benchmark. These results demonstrate the effectiveness of our spatially guided pre-training and action post-training strategies.

## 3.3 EVALUATION IN SIMULATED LARGE-SCALE PICK-AND-PLACE

**Simulation Setup**. Existing benchmarks, such as SimplerEnv and LIBERO, are limited in both scale and diversity, which restricts their capacity to evaluate instruction following manipulation in cluttered and varied environments. To address these limitations, we construct a large-scale simulation benchmark in Isaac-Sim by GenManip Gao et al. (2025), comprising 200 pick-and-place tasks, each involving distinct manipulated objects. With the inclusion of background elements, the benchmark encompasses more than 3,000 objects and containers. Each task was executed once through the data generation pipeline to ensure its executability. Furthermore, for each of the 200 tasks, we additionally collected 5 trajectories with identical object sets but randomized layouts, which were used for post-training. Four evaluation tracks in-distribution, unseen objects, new backgrounds, and unseen instructions were established to assess the model's multidimensional generalization in pick-and-place tasks. Additional details about the evaluation are provided in Appendix Section D.2.

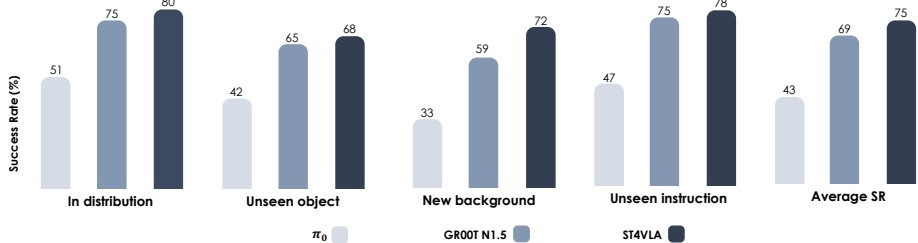

Figure 4: Success rate (%) across different generalization settings on 200 simulated instruction-following pick-and-place tasks.

**Results**. Since both baseline methods, $\pi_0$ Black et al. (2024) and GR00T N1.5 Bjorck et al. (2025), underwent extensive pretraining on large corpora of action data, we ensured a fair comparison by post-training our model on a large-scale dataset of 244K pick-and-place demonstrations simulations generated in Isaac-Sim following stage 2 paradigm. As illustrated in Figure 4, ST4VLA consistently achieves state-of-the-art performance across all four evaluation tracks. Our approach outperforms both $\pi_0$ and GR00T N1.5, highlighting its robust generalization in vision and language, as well as its effectiveness in multi-task learning under spatial guidance.

## 3.4 EVALUATION IN REAL-WORLD CLUTTERED-SCENE PICK-AND-PLACE

We use the Franka Research 3 robot to evaluate the generalization performance of our model and baselines on the real-world pick-and-place tasks. The robot is instructed to sort specified objects into designated containers based on natural language commands. We collect 1K pick-and-place trajectories involving 23 objects and 5 containers, which are used for post-training. Unlike the

200 simulated tasks, the post-training leverages both large-scale simulation data and real-world trajectories. Details of the real-world robot setup and additional experimental configurations are provided in Appendix Section D.3.

Table 4: Comparison of results on real-world generalization of pick-and-place tasks. Success rates (%) are reported. Abbreviations: In dist.: in-distribution; New inst.: new instance; Similar dist.: similar distractors; New bg.: new backgrounds; Unseen obj. pos.: unseen object position; Unseen obj. orient.: unseen object orientation; By attr.: by attribute; By spatial: by spatial relation.

| Models | In dist. | Unseen object | | | Unseen obj. pos. | Unseen obj. orient. | Unseen instruction | | Avg. |
|---|---|---|---|---|---|---|---|---|---|
| | | New inst. | Similar dist. | New bg. | | | By attr. | By spatial | |
| $\pi_0$ Black et al. (2024) | 45 | 32 | 25 | 27 | 18 | 32 | 37 | 31 | 31 |
| GR00T N1.5 Bjorck et al. (2025) | 78 | 46 | 40 | 47 | 20 | 40 | 59 | 53 | 48 |
| **ST4VLA** | **92** | **62** | **49** | **63** | **52** | **72** | **73** | **61** | **65** |

As shown in Table 4, beyond evaluating model performance across multiple tasks in the in-distribution setting, we further assess generalization along four challenging dimensions: unseen objects, unseen object poses and orientations, and novel instructions. ST4VLA outperforms all baselines across real-world test settings. Even under highly challenging conditions, such as interference from visually similar distractors, novel object instances, and paraphrased instructions, ST4VLA achieves strong results through spatial pretraining and spatially guided post-training. These findings demonstrate the model's robust visual and linguistic generalization in pick-and-place tasks. Furthermore, in evaluations involving unseen object poses and orientations, our approach significantly surpasses the baselines $\pi_0$ and GR00T N1.5, benefiting from the diverse grasp positions and trajectories introduced by co-training on large-scale simulation data.

## 3.5 EVALUATION IN LONG-HORIZON MANIPULATION

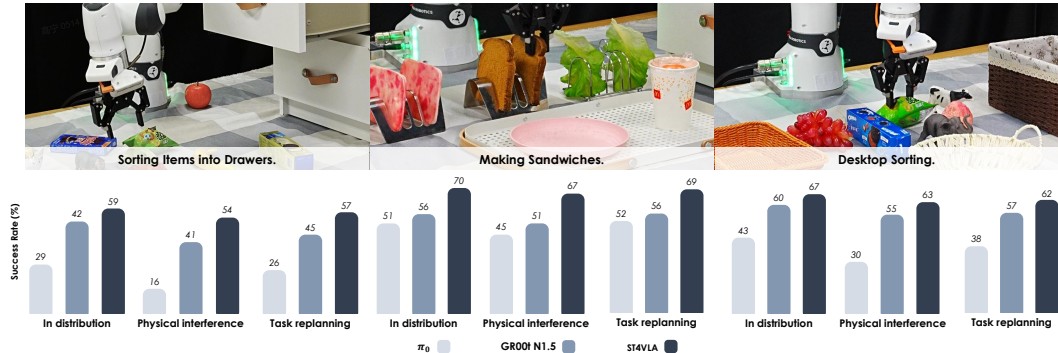

Figure 5: Demonstration and results of long-horizon instruction-following manipulation tasks.

A key strength of our dual-system framework is its ability to leverage the high-level planner System 2 to decompose complex, reasoning-heavy tasks into sequences of atomic actions, which are then robustly executed by the low-level controller System 1. To evaluate this capability, we design tasks such as desktop sorting, drawer organization, sandwich making, requiring multi-step planning, progress monitoring, and dynamic adaptation. We collect 22 hours of teleoperated demonstrations, segment trajectories into subtasks, and train ST4VLA jointly on task decomposition, subtask identification and action prediction. Performance is evaluated under three settings: In-distribution, physical interference and task replanning. Results in Figure 5 show that ST4VLA consistently surpasses GR00T N1.5 and $\pi_0$, reliably grounding high-level goals into executable steps, adapting to disturbances, and dynamically revising plans with minimal degradation, demonstrating strong resilience in dynamic, real-world environments. Additional details on the long-horizon task setup and evaluation settings are provided in Appendix Section D.4.

## 4 RELATED WORK

**Hierarchical robot system.** Bridging high-level instructions with low-level actions is a central challenge in embodied AI, often addressed by introducing intermediate representations (IRs) ranging from

symbolic structures to learned embeddings Xie et al. (2019). Inspired by Chain-of-Thought reasoning, many works train vision-language-action (VLA) models to first output textual plans, improving interpretability and long-horizon performance Zawalski et al. (2024). Beyond text, IRs have taken the form of perceptual cues (e.g., bounding boxes Griffin (2023), grasp points Ten Pas & Platt (2017), or dense features Laskin et al. (2020); Nair et al. (2022)), persistent 3D scene graphs for grounding plans Rana et al. (2023), and action-centric affordances specifying end-effector poses Nasiriany et al. (2024). Recent work further generates spatial localizers directly usable by controllers Huang et al. (2025a); Gu et al. (2023); Li et al. (2025c), or leverages large foundation models that unify planning with affordance prediction Team et al. (2025); Luo et al. (2025). Specialized models such as RoboRefer Zhou et al. (2025a) target fine-grained spatial grounding with reinforcement learning. LLaRA Li et al. (2024d) similarly adapts VLMs to robotic control via instruction-style data. In contrast, our method unifies these directions by modeling latent spatial guidance jointly with action learning, enabling end-to-end optimization from real-world feedback.

**Embodied reasoning and planning in VLA.** Several recent VLA approaches leverage large-scale multimodal co-training to improve generalization. RT-2 Brohan et al. (2023), ChatVLA Zhou et al. (2025c), and GR-2/3 Cheang et al. (2024; 2025) combine internet vision–language data with robot trajectories, while InstructVLA Yang et al. (2025b) and $\pi_{0.5}$ Intelligence et al. (2025) further incorporate instructional signals, sometimes with spatial annotations such as bounding boxes, to enhance language–action alignment. Parallel efforts explore explicit reasoning: ECOT Zawalski et al. (2024) generates textual plans, RT-H Belkhale et al. (2024) introduces action language for hierarchical control, InstructVLA Yang et al. (2025b) jointly optimizes reasoning and action, OneTwoVLA Lin et al. (2025) alternates between "thinking" and execution, RAD Clark et al. (2025) distills reasoning from human videos, and graph-based IRs Huang et al. (2025b) support spatial reasoning. Beyond textual reasoning, recent works have incorporated visual foresight and structural planning: CoT-VLA Zhao et al. (2025) generates future video frames as a visual chain-of-thought, Chain-of-Action Zhang et al. (2025) and LBP Liu et al. (2025a) apply backward goal-based planning, while ATM Wen et al. (2023) extracts control signals from unlabeled videos via point-trajectory prediction and LLARVA Niu et al. (2024) leverages visual-trace representations to align vision and action. While these methods expand semantics and interpretability, they often treat multimodal data as generic supervision, overlook explicit spatial grounding, and rely on costly generative reasoning. In contrast, our approach strategically emphasizes spatial grounding data and introduces a spatially guided co-training scheme with gradient alignment, coupled with a lightweight post-training phase that unlocks intrinsic reasoning in VLMs without requiring explicit outputs.

**Generalist robot policy.** Recent progress in general-purpose robotics follows three main paradigms. Monolithic VLA models directly map multimodal inputs to tokenized actions with a single network Brohan et al. (2023); Kim et al. (2024); Lee et al. (2025). Unified architectures decouple high-level cognition from low-level control, enabling modularity and interpretability Black et al. (2024); Li et al. (2024c; 2025a); Zheng et al. (2024); Intelligence et al. (2025); Song et al. (2025); Zhou et al. (2025b); Yang et al. (2025b); Shukor et al. (2025); Cheang et al. (2025). World models instead learn predictive environment dynamics to plan in latent space, offering strong foresight but at higher computational cost Ye et al. (2025b); Bjorck et al. (2025); Li et al. (2025b); Cen et al. (2025); Liao et al. (2025); Tian et al. (2024); Bu et al. (2025b); Wang et al. (2025); Lv et al. (2025). Similar to ours, Magma Yang et al. (2025a) also adopts spatial pre-training, though it does not explicitly leverage spatial prompting to guide action generation. Our approach extends unified VLA architectures with a dual-system design that boosts adaptability for real-world tasks.

## 5 DISCUSSION AND CONCLUSION

In this work, we presented ST4VLA, a unified vision-language-action framework that leverages spatial grounding priors to bridge high-level multimodal reasoning with low-level robotic execution. By combining large-scale multimodal pre-training with spatially guided post-training, our model effectively transfers perceptual and reasoning skills into embodied control, achieving strong generalization to unseen objects, instructions, and environments. Extensive evaluations across simulation and real-world settings demonstrate that ST4VLA surpasses existing VLA models and specialized systems in instruction following, long-horizon manipulation, and multimodal grounding, highlighting spatial reasoning as a unifying substrate for scalable and reliable generalist robots.

## ACKNOWLEDGMENTS

This research is supported by Shanghai Artificial Intelligence Laboratory, and is funded in part by the National Key R&D Program of China (2022ZD0160201). Jinhui Ye is partly supported by the Research Grants Council under the Areas of Excellence scheme grant AoE/E-601/22-R. Prof. Yanwei Fu is partly supported by Shanghai Municipal Science and Technology Major Project (2025SHZDZX025G02). This paper systematically analyzes the impact of spatial training on Vision-Language-Action (VLA) models and extends the manipulation of spatial data for generalist purposes within the InternVLA-M1 framework Contributors (2025). We would like to extend our sincere gratitude to all contributors for their work on InternVLA-M1 Team and the Intern Robotics Team, which encompasses data collection, model development, simulation, benchmarking, real-robot deployment, and open-source efforts: `Xinyi Chen`, `Jiaya Jia`, `Hao Li`, `Yao Mu`, `Yu Qiao`, `Yang Tian`, `Bolun Wang`, `Hanqing Wang`, `Tai Wang`, `Ziqin Wang`, `Xueyuan Wei`, `Chao Wu`, `Shuai Yang`, `Jia Zeng`, `Jingjing Zhang`, `Shi Zhang`, `Bowen Zhou` (*listed in alphabetical order by last name*).

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

# Supplementary Materials
# ST4VLA: Spatially Guided Training for Vision-Language-Action Model

## Table of Contents

## A    PROJECTION-SPACE SIMILARITY (PSS)

**Setup.**    To quantify the alignment between the optimization directions of spatial grounding and robot manipulation tasks, we analyze the similarity of their loss gradients with respect to the shared parameters. Let $\theta \in \mathbb{R}^{d \times n}$ denote the shared parameters of the model (i.e., the VLM backbone). We fix two probing mini-batches: a batch of grounding data $\mathcal{B}\text{spat}$ and a batch of action data $\mathcal{B}\text{act}$. We then compute the gradients of the respective losses with respect to $\theta$ for each batch, resulting in the following gradient matrices:

$$G_{\text{spat}} = \nabla_\theta \mathcal{L}\text{spat}(\mathcal{B}\text{spat}; \theta) \in \mathbb{R}^{d \times n}, \quad G_{\text{act}} = \nabla_\theta \mathcal{L}\text{act}(\mathcal{B}\text{act}; \theta) \in \mathbb{R}^{d \times n}. \tag{1}$$

**Projection-space similarity (PSS).**    To capture structural alignment between the spatial grounding objective and the action manipulation objective, we compare the subspaces spanned by $G_{\text{spat}}$ and $G_{\text{act}}$ via Singular Value Decomposition (SVD). Let $P_{\text{spat}}$ and $P_{\text{act}}$ be the orthogonal projectors onto $\text{range}(G_{\text{spat}})$ and $\text{range}(G_{\text{act}})$, respectively. Using the Moore–Penrose pseudoinverse $(\cdot)^+$,

$$P_{\text{spat}} = G_{\text{spat}} G_{\text{spat}}^+, \qquad P_{\text{act}} = G_{\text{act}} G_{\text{act}}^+. \tag{2}$$

Denote $r_{\text{spat}} = \text{rank}(G_{\text{spat}})$ and $r_{\text{act}} = \text{rank}(G_{\text{act}})$. The projection-space similarity is define as:

$$\text{PSS}(G_{\text{spat}}, G_{\text{act}}) = \frac{\text{tr}(P_{\text{spat}} P_{\text{act}})}{\min(r_{\text{spat}}, r_{\text{act}})} \in [0, 1], \tag{3}$$

which equals the mean of squared cosines of the principal angles between the two subspaces. A value of 1 indicates identical subspaces.

**Protocol.**    Given the billion-scale parameters of the VLM backbone, computing gradients for the entire model would be computationally prohibitive. Therefore, we restrict our analysis to a single layer: the $q$ projection in the self-attention module of the *final layer* of the Qwen language model, whose parameter $\theta \in \mathbb{R}^d$ is a $2048 \times 2048$ weight matrix. We focus on this particular layer because it lies at the interface between the language model backbone and the action expert, making it the most informative point for capturing the interaction between the two components.

During training, we periodically compute PSS using the fixed probing evaluation sets (batch size = 64 for each type of data). A higher PSS indicates that the action policy optimization is well-aligned with the features learned through multimodal grounding.

## B    ADDITIONAL EXPERIMENTS

### B.1    FURTHER STUDY FOR SPATIAL PROMPTING

To address the question of whether spatial priors merely accelerate convergence or are essential for final performance capability, we extended the training horizon of all models to 100k steps. As hypothesized, standard baselines might require longer training to fully saturate; however, our extended analysis demonstrates that the performance gap remains significant even after convergence.

Figure 6 illustrates the training dynamics across 100k steps for both WidowX and Google Robot environments. While the *Vanilla VLA* and *Vanilla Co-training* baselines continue to show marginal improvements early on, they ultimately converge to substantially lower plateaus compared to our method. These results conclusively show that explicitly injecting spatial priors does not simply act as a warm-up for faster learning; it fundamentally alters the optimization landscape, allowing the policy to generalize to a higher performance ceiling that standard multimodal co-training fails to reach.

### B.2    LIBERO BENCHMARK

**LIBERO**. LIBERO is a language-conditioned manipulation suite built on a Franka arm with diverse scenes and expert demonstrations. We evaluate four task sets: LIBERO-Spatial (same objects, different spatial layouts), LIBERO-Object (fixed layout, different objects), LIBERO-Goal (fixed objects and layout, different goals), and LIBERO-Long (also known as LIBERO-10; longer tasks that span multiple objects, layouts, and operations).

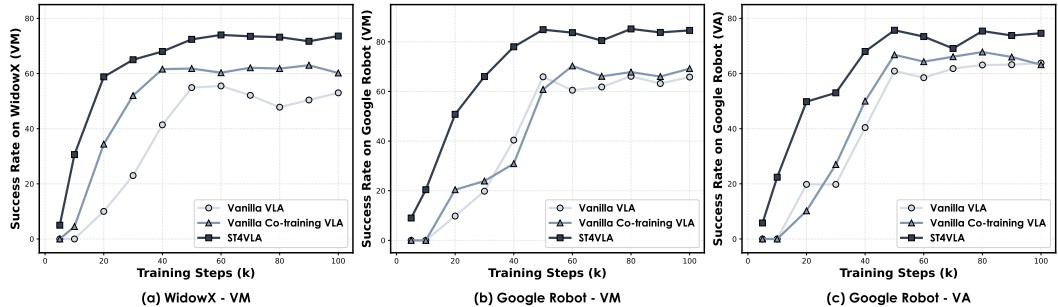

Figure 6: Extended training curves up to 100k steps on WidowX and Google Robot tasks. Even with prolonged training, baselines saturate at a significantly lower performance level compared to our method (ST4VLA), confirming that spatial grounding improves the policy's upper bound rather than just convergence speed.

Table 5: Result comparisons of robotic manipulation on LIBERO (Franka) benchmark.

| Models | Spatial | Objects | Goal | Long | Avg |
|---|---|---|---|---|---|
| OpenVLA Kim et al. (2024) | 84.7 | 88.4 | 79.2 | 53.7 | 76.5 |
| SpatialVLA Qu et al. (2025) | 88.2 | 89.9 | 78.6 | 55.5 | 78.1 |
| CoT-VLA Zhao et al. (2025) | 87.5 | 91.6 | 87.6 | 69.0 | 83.9 |
| GR00T N1 Bjorck et al. (2025) | 94.4 | 97.6 | 93.0 | 90.6 | 93.9 |
| $\pi_0$ Black et al. (2024) | 96.8 | 98.8 | 95.8 | 85.2 | 94.2 |
| $\pi_0$-FAST Pertsch et al. (2025) | 96.4 | 96.8 | 88.6 | 60.2 | 85.5 |
| $\pi_{0.5}$-KI Driess et al. (2025) | 98.0 | 97.8 | 95.6 | 85.8 | 94.3 |
| Vanilla VLA | **98.8** | 98.0 | 81.4 | 88.0 | 91.6 |
| **ST4VLA** | 98.0 | **99.0** | **93.8** | **92.6** | **95.9** |

**Experimental setups.** Following Kim et al. (2025), we filter out failed demonstrations and pause frames. During training, the policy takes as input both wrist-mounted and third-person camera views. We fine-tune the model on each suite independently using 8 A100 GPUs with a batch size of 128 and an action chunk size of 8. Training runs for roughly 30K steps, lasting about 20 hours. Each suite is evaluated with 500 trials.

**Result analysis.** The primary experimental results on the LIBERO benchmark are presented in Table 5. Compared to previous strong baselines, such as GR00T N1 and $\pi_0$, the ST4VLA framework achieves notable improvements, particularly on the spatial and long-horizon tracks, with success rates of 98.0% and 92.6%, respectively. These results demonstrate the efficacy of our proposed method in managing complex, multi-step manipulation tasks. Specifically, for object placement, ST4VLA attains a 99.0% SR, which highlights its robust object grounding capability.

## C  ABLATION STUDIES

As described in Section 3, ST4VLA achieves significant performance improvements on the SimplerEnv benchmark. In this section, we conduct ablation studies to examine the contribution of each critical component. Specifically, we investigate: (1) the impact of different pretrained models used in the first-stage pre-training; (2) the ratio of multimodal data during the second-stage post-training; (3) whether incorporating spatial prompts into task instructions brings improvement.

### C.1  IMPACT OF SPATIAL GROUNDING PRE-TRAINING

We first analyze the impact of different pre-training data configurations in the first-stage pre-training. Our model is evaluated under three settings: 1) Using the official `QwenVL-2.5-3B-Instruct` weights without additional spatial pre-training; 2) Pretraining with general multimodal grounding

data (e.g., LLaVA-OneVision and RefCOCO); 3) Pretraining with ST4VLA robotic grounding data. The proportions of each data type used are provided in the Appendix Section F.

Table 6: Performance comparison under different pretraining data settings.

| Pretraining Data | Robotic Grounding (pre-training) | | | Robotic Manipulation | |
|---|---|---|---|---|---|
| | Where2place Acc | Refit-testB IoU@0.5 | A0 Maniskill L2 Dist. | Google Robot VM/VA | WidowX VM |
| No Additional Pretraining | 0 | 69.0 | - | 66.1/63.5 | 54.9 |
| + General Grounding Data | 30.7 | 74.9 | - | 72.6/70.3 | 65.2 |
| + Robotic Grounding Data | 60.5 | 83.4 | 3.6 | 84.3/75.9 | 73.1 |

**Result analysis.** We assessed the pre-training Vision-Language Model (VLM) using the Grounding dataset for grounding performance, Where2Place Yuan et al. (2024) for point prediction, RoboRefit Lu et al. (2023) for bounding box prediction, and A0 ManiSkill Xu et al. (2025b) for trajectory prediction. As shown in the upper section of Table 6, the base model `QwenVL-2.5-3B-Instruct` demonstrates the ability to detect bounding boxes for operation-related objects, but struggles to accurately point objects or predict trajectories. Despite these limitations, a Vanilla VLA built upon this model still achieves competitive performance in SimmerEnv compared to $\pi_0$ (e.g., 54.9 vs. 48.3). By pretraining the VLM with open-source multimodal grounding data (e.g., RefCOCO), we observe improved object recognition capabilities, specifically an increase in Box IoU@0.5 from 69.0 to 74.9. This enhancement leads to a significant performance gain on the WidowX benchmark (54.9 $\rightarrow$ 65.2), demonstrating that visual grounding pretraining effectively improves downstream manipulation accuracy.

Furthermore, incorporating robotic grounding data such as ST4VLA equips the VLM with the ability to interpret point, box, and trajectory keypoints for object interaction. These advancements contribute to the state-of-the-art performance of ST4VLA on SimmerEnv, including a 12.4% improvement on the WidowX benchmark.

## C.2 THE IMPACT OF COTRAIN LOSS WEIGHT

In this section, we examine the influence of cotraining strategies during the post-training stage. We find that while cotraining significantly affects model performance, the ratio between robotic loss and multimodal loss plays a crucial role. We ablate different loss mixing ratios introduced in post-training and summarize the results in Table 7.

Table 7: Performance comparison under different pretraining data settings. *Input image resized to 224×224 to align with prior work Black et al. (2024); Kim et al. (2024; 2025).

| Loss weight ratio (grounding vs. action) | Spatial grounding | | | Robotic Manipulation | |
|---|---|---|---|---|---|
| | Where2place Point-Acc | Refit-testB IoU@0.5 | A0 Maniskill MAE Dist. | Google Robot VM/VA | WidowX VM |
| 1:1 | 50.3 | 80.2 | 3.5 | 52.4/42.4 | 47.2 |
| 1:5 | 48.3 | 80.0 | 4.0 | 63.8/52.5 | 58.3 |
| 1:10 | 42.3 | 80.4 | 5.5 | 80.7/76.0 | 71.7 |
| 1:15 | 38.5 | 75.8 | 5.6 | 80.7/70.2 | 71.8 |
| 1:20 | 31.3 | 74.1 | 6.0 | 78.3/65.2 | 68.3 |

**Results analysis.** The results indicate that cotraining ratios such as 1:1 or 1:5 can further enhance the VLM's robotic grounding capability, but lead to a considerable decline in manipulation performance (e.g., 73.2 $\rightarrow$ 47.2). However, when the cotraining ratio is increased to 1:15 or 1:20, the manipulation performance also declines. This indicates that the relationship between multimodal cotraining and manipulation is not a simple trade-off. The optimal ratio is observed to be 1:10. We hypothesize that this ratio corresponds approximately to the proportion between the action chunk length and the average next-token prediction length in the multimodal data.

## C.3 BACKBONE-AGNOSTIC GENERALIZATION AND TRAINING METHOD CONTRIBUTION

To assess whether the effectiveness of our proposed training framework depends on the capacity of the underlying VLM backbone, we conducted two complementary evaluations. First, we rebuilt ST4VLA using Florence-2 Xiao et al. (2024), a considerably weaker VLM compared to Qwen2.5-VL or the GR00T backbone, and compared it against GR00T N1.5 and a Vanilla Co-training baseline. Second, to isolate the contribution of our spatial grounding training stage from backbone capacity, we performed controlled ablations where all models share the identical Qwen2.5-VL-3B backbone.

Table 8: Comparison across different VLM backbones (Florence-2 vs. Qwen2.5-VL-3B) and training methods.

| Backbone | Model | Put Spoon on Towel | Put Carrot on Plate | Stack Green on Yellow | Put Eggplant in Basket | Average |
|---|---|---|---|---|---|---|
| Eagle-2.5 | GR00T N1.5 | 75.3 | 54.3 | **57.0** | 61.3 | 61.9 |
| Florence-2 | Vanilla Co-training VLA | 75.2 | 31.3 | 3.1 | 75.0 | 46.1 |
| | **ST4VLA** | 79.6 | 70.5 | 28.3 | 93.0 | 67.9 |
| Qwen2.5-VL-3B | Vanilla VLA | 56.6 | 63.3 | 27.0 | 71.8 | 54.7 |
| | Vanilla Co-training VLA | 70.3 | 68.4 | 20.5 | 85.2 | 61.1 |
| | **ST4VLA** | **80.2** | **79.2** | 35.4 | **98.0** | **73.2** |

**Result analysis.** Across both settings, as shown in Table 8, the evidence consistently shows that the benefits of ST4VLA do not stem from backbone capacity. With a much weaker Florence-2 backbone, ST4VLA still surpasses GR00T N1.5 (67.9% vs. 61.9%), while the Vanilla Co-training baseline collapses on difficult tasks (e.g., 3.1% for Block Stacking). Under controlled conditions with identical Qwen2.5-VL-3B backbones, A clear improvement trajectory is observed: Vanilla VLA achieves 54.7, Vanilla Co-training reaches 61.1, and ST4VLA further improves to 73.2. This progression confirms that the performance gains arise from our spatial grounding training stage rather than from the backbone capacity.

## C.4 SCALING LAWS OF SPATIAL PRIORS

Finally, we investigated the scaling behavior of spatial priors by varying the Spatial Grounding Pre-training data volume from 0M to 3M pairs, followed by standard Post-training on OXE.

Table 9: Ablation on the scaling of Spatial Grounding Pre-training data volume.

| Pre-training Scale | Google Robot VM | Google Robot VA | WidowX VM | Average |
|---|---|---|---|---|
| 0 M | 66.1 | 63.5 | 54.7 | 61.4 |
| 0.5 M | 66.1 | 61.2 | 55.6 | 61.0 |
| 1.0 M | 68.9 | 65.5 | 55.8 | 63.4 |
| 2.0 M | 72.8 | 72.9 | 67.3 | 71.0 |
| 3.0 M | **84.6** | **75.9** | **73.2** | **77.9** |

**Result analysis.** Our experimental results, shown in Table 9, reveal a nonlinear relationship between spatial data scale and model performance. Performance gains remain modest when spatial grounding data is below 1.0M pairs. However, once the data scale surpasses 2.0M pairs, we observe dramatically increasing returns. At 3.0M pairs, the model achieves substantial improvement, with average performance rising from 61.4 to 77.9—a remarkable 26.9% relative gain. This suggests that a critical mass of spatial grounding data is required to unlock the VLM's full manipulation potential.

## C.5 ABLATION STUDY ON SPATIAL PROMPT FORMULATIONS

In our default implementation, ST4VLA employs a **single unified spatial prompt** across all tasks: *"Figure out how to execute it, then locate the key object needed."* This design encourages the model to

attend to spatial features without strictly enforcing a specific output format (e.g., bounding boxes or points) during the action prediction phase.

To investigate whether the specific phrasing or the imposition of explicit spatial constraints influences manipulation performance, we conducted an ablation study comparing our unified prompt against the following four variants:

- **Unified Prompting (Default):** *"Figure out how to execute it, then locate the key object needed."*
- **Random Padding:** Uses a non-semantic sequence to test if gains are due to sequence length alone: *"xxx, xxx, xxx, xxx, xxx, xxx"*.
- **Box Prompting:** Appends a specific constraint demanding bounding box coordinates: *"Figure out how to execute it, then locate the key object needed. Give the box coordinates according to the instruction"*.
- **Point Prompting:** Appends a specific constraint demanding a list of tuples for points: *"Figure out how to execute it, then locate the key object needed. Your answer should be formatted as a list of tuples"*.
- **Trace Prompting:** Appends a constraint demanding a trajectory prediction: *"Figure out how to execute it, then locate the key object needed. Based on the task description predict the trajectory that the end effector should take"*.

Table 10: Ablation analysis of different spatial prompt formulations on SimplerEnv, comparing the default Unified Prompt against non-semantic and explicit formatting constraints.

| Prompt Type | Google Robot VM | Google Robot VA | WidowX VM | Average |
|---|---|---|---|---|
| Random Padding | 64.2 | 60.8 | 50.6 | 58.5 |
| **Unified Prompting (Ours)** | **84.6** | **75.9** | 73.2 | **77.9** |
| Box Prompting | 80.9 | 73.0 | **75.8** | 76.6 |
| Point Prompting | 80.8 | 70.7 | 73.3 | 74.9 |
| Trace Prompting | 79.6 | 70.9 | 71.2 | 73.9 |

**Result analysis.** The results in Table 10 provide two key insights:

1. **Semantic content matters:** The *Random Padding* baseline significantly underperforms the *Unified Prompting* (58.5% vs. 77.9%). This confirms that the performance gains in ST4VLA stem from the model explicitly attending to spatial semantics, rather than merely from the computational overhead of processing extra tokens.

2. **Unified prompting is sufficient:** Our default *Unified Prompting* achieves the highest average success rate (77.9%), outperforming variants that enforce strict output constraints like Box (76.6%), Point (74.9%), or Trace (73.9%). This suggests that while spatial awareness is critical, rigidly forcing the VLM to format its internal reasoning into specific coordinates during action inference is unnecessary and may even slightly constrain the policy's flexibility.

Therefore, our unified prompt serves as an optimal, task-agnostic instruction that robustly activates spatial attention across diverse manipulation tasks.

## D  EXPERIMENTS SETUP

### D.1  EVALUATION IN SIMPLERENV

**Experiment Setup.** As described in Section 2.2, we post-train ST4VLA on a subset of Open-X Embodiment (OXE) (including `fractal_rt_1` and `bridge_v1`), with co-training on spatial grounding data (Figure 1). The VLM takes the primary observation image, task instruction, and an auxiliary spatial prompt as input, while the action expert predicts actions with an action chunk size of 16. For multimodal data, the model follows an SFT-style question-answering format. Training is performed on 16 NVIDIA A100 GPUs for 50k steps (~2.5 epochs), with batch sizes of 16 for robot action data and 4 for multimodal data, optimized with a summed loss over both data types. All evaluations are conducted within SimplerEnv using its official evaluation protocol.

## D.2 EVALUATION IN SIMULATED LARGE-SCALE PICK-AND-PLACE

**Evaluation settings.** As illustrated in Figure 7, our evaluation consists of four distinct tracks: (1) *In-distribution*, which evaluates the model's pick-and-place capability on identical object instances under varied layouts corresponding to post-training scenarios; (2) *Unseen object*, where in each of the 200 scenes the graspable target object is replaced with one not encountered during training, thereby testing the model's generalization to novel object instances; (3) *New background*, in which the table and background textures of the in-distribution scenes are altered to assess visual robustness; (4) *Unseen instruction*, where the original template instruction "Move obj1 to the top of container1" is reformulated using GPT-4o-mini, using prompt as shown in the box below,, introducing variations in object attributes and grammatical structures to evaluate the model's capacity to generalize to novel linguistic commands. In these tasks, the graspable target object is randomly placed within a $20 \times 35$ cm region in front of the robot base, while the container is randomly positioned within a $40 \times 70$ cm area. Nine additional background objects are scattered across the tabletop at random. A data generation pipeline constructs each testing layout, ensuring that every configuration remains solvable for successful grasping and placement. Each track includes 200 distinct scenes, with a maximum of 600 steps permitted per trial. A trial is deemed successful if the object is placed atop the designated container within the step limit.

**Experiment setup.** The observation space consists of two RGB images: one captured from a fixed third-person viewpoint and the other from a first-person camera mounted on the Franka end-effector. Both images are resized to $224 \times 224$ before being input to the model. For comparison, the baseline methods additionally incorporate a 7-dimensional representation of the Franka joint states. The VLA model outputs an 8-dimensional continuous action vector, where 7-dimensions correspond to the incremental deltas of each joint and one dimension encodes the binary signal for gripper control. Each action vector has a temporal chunk size of 16 and, after temporal ensembling, is applied to the Franka robot in the simulation environment.

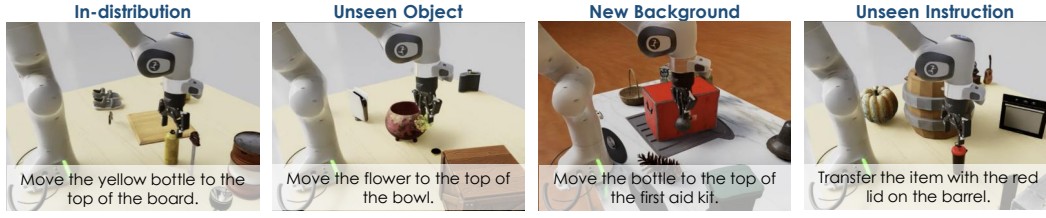

Figure 7: Evaluation settings for generalizable pick-and-place in large-scale simulation.

## D.3 REAL-WORLD PICK-AND-PLACE MANIPULATION SETUP

**Evaluation settings.** To evaluate generalization, we divide all available object and container assets into disjoint *seen* and *unseen* sets, as illustrated in Figure 8. The training phase uses only the seen set, while testing incorporates both sets to assess the model's ability to handle novel objects. We examine real-world pick-and-place generalization across several conditions: in-distribution, unseen object, unseen object position, unseen object orientation, and unseen instruction. Among these, the unseen instruction and unseen object settings (depicted in Figure 9) introduce complementary reasoning challenges. The unseen instruction setting involves two key reasoning types, namely spatial reasoning and attribute identification, whereas the unseen object setting encompasses three categories, including new object instances, similar distractors, and new backgrounds. (1) **Spatial reasoning**, where the robot must act based on relative spatial relationships (e.g., "Place the object closest to the robot base into the brown box"); (2) **Attribute identification**, which requires grounding instructions in specific visual attributes such as color or shape (e.g., "Move the green fruit into the white fruit plate"); (3) **New object instances**, where novel objects not encountered during training must be manipulated (e.g., "Put the small chips into the brown fruit plate"); (4) **Similar distractors**, which probe the model's ability to disambiguate between nearly identical objects (e.g., distinguishing a blue Oreo from other cookies); (5) **New backgrounds**, where the robot must adapt to altered visual contexts while grounding instructions (e.g., "Move the pear onto the pink fruit plate"). Together, these variations establish a comprehensive and challenging benchmark that evaluates instruction

following across perception, reasoning, and generalization. In contrast, the unseen object position and unseen object orientation settings involve seen objects whose grasping or placing positions, as well as orientations, are shifted during testing, such that they differ from the regions or rotational angles encountered during training.

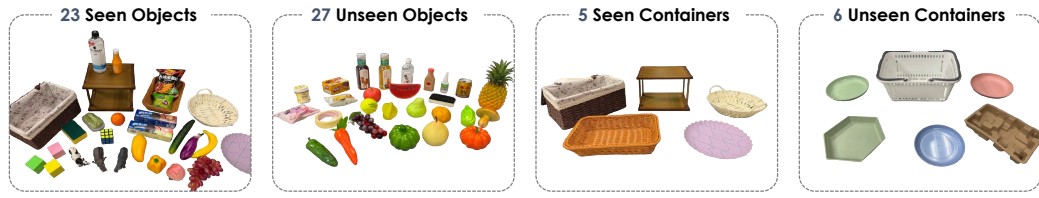

Figure 8: Overview of objects and containers used in instruction-following pick-and-place.

For each model, we conduct a total of 300 rollout evaluations. Each trial may correspond to one or more testing settings, and we ensure that each setting is evaluated at least 50 times. Each trial allows up to three consecutive attempts. For each trial, three containers are chosen and placed at fixed tabletop locations within a $60 \times 90$ cm workspace, and a larger collection of objects is randomly scattered between them. This configuration ensures that the robot must rely on precise perception and instruction grounding, rather than memorized placements, to correctly execute the instructed pick-and-place actions. We report the success rate (SR), defined as the fraction of trials in which the specified object is successfully placed into the designated container. A higher SR indicates better performance. To ensure fair comparisons across models, we fix the positions of the objects and containers for each task during testing.

**Experimental setup.** We collected six hours of teleoperated demonstration data with seen objects and containers to serve as post-training real-world data. The two RGB views were resized to $224 \times 224$ and used as model inputs. For comparison, the baseline methods additionally incorporate a 6-dimensional representation of the end-effector's position and orientation. The VLA model outputs a 7-dimensional continuous action vector: six dimensions correspond to the incremental deltas of the end-effector's position and orientation, and one dimension encodes the binary signal for gripper control. Each action vector is organized into a temporal chunk of size 16, which, after temporal ensembling, is applied during execution.

---

**Prompt Description: Attribute-based Instruction Rewriting**

**Task Overview.** The task is to optimize an existing instruction for a robot model to enhance attribute-specific grounding.

**Input:**
- **Pick obj description:** Object description obtained from simulation (e.g., *red apple*).
- **Container description:** Container description obtained from simulation.
- **Raw Instruction:** Original instruction (e.g., *Move obj1 to the top of container*).
- **Rewrite guideline:** Focus on materials, color, or shape.

**Rules:**

1. There are many items on the desktop. Ensure the rewritten instruction is specific enough to unambiguously identify the target object and container.

2. If the attributes mentioned in the original instruction (such as shape, color, or material) are not sufficient to uniquely identify the object, add extra features (e.g., relative position, size, or additional visual properties) to remove ambiguity.

3. Do **not** mention the object's common name (e.g., do not say "apple" or "cup").

4. The rewritten instruction must sound natural and fluent while preserving the original meaning.

**Output:**
- Provide **5 examples** of optimized instructions in JSON format (a list of strings).
- Keep all examples simple, clear, and easy to understand.

**Example:**

```
input: Place the apple to the top of plate.
output: [
  "Put the red sphere on top of the round white plate.",
  "Stack the small red object onto the large white plate.",
  "Set the red fruit on the circular plate.",
  "Position the shiny red sphere on top of the white ceramic plate.",
  "Move the red object closest to the center onto the round plate."
]
```

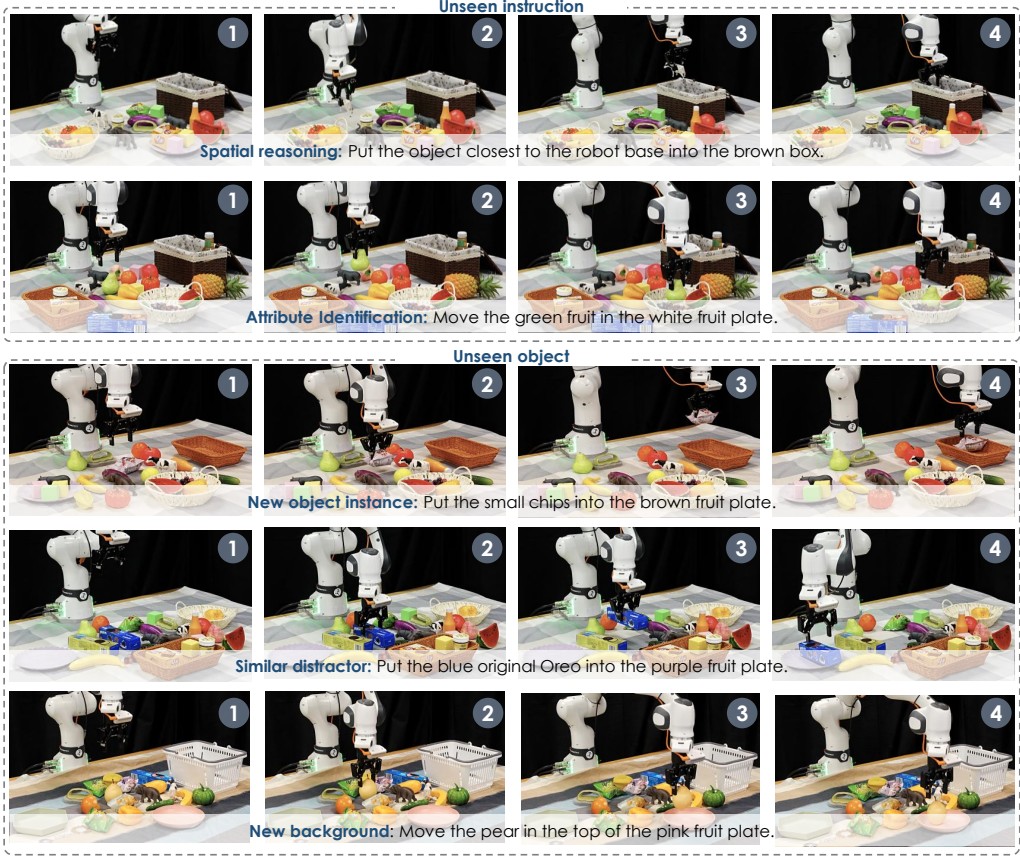

Figure 9: Evaluation settings showcases for real-world generalization pick-and-place.

## D.4 REAL-WORLD LONG-HORIZON MANIPULATION SETUP

**Evaluation settings.** We evaluate model performance under three distinct settings: in-distribution, physical interference and task replanning:

- **Physical interference.** External disturbances are introduced during task execution. For example, during the *sorting items into drawers* task, the drawer is manually closed after the robot opens it, or the target object is displaced during grasping. This evaluates the model's ability to perceive environmental changes and adapt accordingly.
- **Task replanning.** New instructions are issued mid-execution. For instance, after placing an object in the drawer but before closing it, the robot is told: "Also put the cow toy into the top drawer." This tests the model's ability to incorporate new subgoals and dynamically adjust its plan.

The tasks illustrated in Figure 10 include:

- **Desktop sorting.** The Franka robot is tasked with sorting objects into containers based on high-level semantic categories, aiming to ensure that all items on the desktop are eventually placed into the correct containers. Both objects and containers are scattered within a 60×90 cm region in front of the robot base. The setup includes five seen containers and five object categories: *fruits, toys, vegetables, bottles*, and *snacks*. Each evaluation instance requires sorting objects from one to three categories into their designated containers, with each trial comprising three sequential pick-and-place actions. For every method, evaluations are conducted more than 30 times across the three settings, ensuring that each individual setting is tested at least 10 times. A success is recorded upon the completion of each individual pick-and-place operation, and we report the final overall success rate accordingly.

- **Sorting items into drawers.** The Franka robot is required to (i) open a designated drawer (either lower or upper), (ii) place the target objects into it, and (iii) close the drawer. This task demands precise temporal reasoning and articulated manipulation. The objects are placed within a 35×35 cm area located to the front-right of the robot base. As in the previous setting, the number of trials remains the same; however, here we report stepwise execution success, where a step is deemed valid only if all preceding steps have been successfully completed.
- **Making sandwiches.** The Franka robot is instructed to assemble sandwiches following a pre-defined meal recipe. Ingredients and plates are placed within a 50×70 cm region in front of the robot base. We define five types of sandwich recipes as the seen set: [ bread–lettuce–bread ], [ bread–lettuce–meat–bread ], [ bread–meat–lettuce–meat–bread ], [ bread–meat–meat–bread ], and [ bread–meat–bread ]. We report success rates on both the seen set and an unseen set involving real-time environment interaction, using the same success definition as in the drawer sorting task.
- **Math calculation.** The Franka robot is prompted to solve a math problem and press the color-coded button (red, yellow, or blue) that corresponds to the correct answer based on arithmetic reasoning. The buttons are randomly placed within a 40×40 cm area in front of the robot base.
- **Goods purchase.** The ARX LIFT2 dual-arm robot is tasked with identifying and placing into a basket the object bearing the correct price tag, given a numerical cue ranging from 1 to 9. We report the success rate of correctly placing the item corresponding to the queried price into the basket.

**Experimental setup.** We collected 22 hours of teleoperated demonstrations (400–500 per task) for long-horizon training, segmenting trajectories into *subtasks* with atomic actions. We introduce zero-action vectors padding after each subtask segment. This allows the model to stop upon subtask completion and then be prompted to predict the transition to the next subtask. Unlike prior VLA models relying on external planners, ST4VLA jointly trains on multimodal inputs, task decomposition, subtask identification, numerical reasoning, and action supervision, for unified planning and action prediction.

### D.5 REAL-WORLD ROBOT SETUP

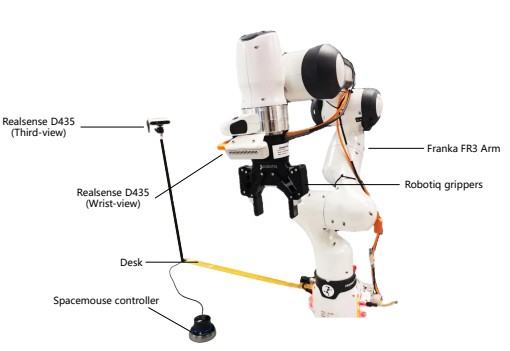
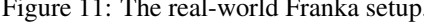

Figure 11: The real-world Franka setup.

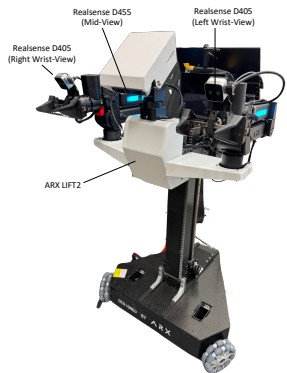

Figure 12: The Real-world ARX LIFT2 setup.

As illustrated in Figure 11, we utilize a Franka Research 3 robot equipped with a Robotiq-2F-85 gripper to evaluate real-world tasks, including short-range pick-and-place, long-horizon object sorting, opening and closing a drawer, and making sandwiches. In our experimental setup, two RealSense D435 cameras capture RGB images for visual input: one is positioned at a rear-side, third-person perspective, and the other is mounted on the Franka's end-effector. Furthermore, as shown in Figure 12, we conduct pick-and-place experiments in shopping scenarios using the dual-arm ARX LIFT2 platform. Each arm is equipped with a RealSense D405, while a RealSense D455 is mounted on the head to capture RGB imagery from a frontal viewpoint. All model inferences are executed on a workstation powered by an NVIDIA RTX 4080 GPU with 16 GB of VRAM.

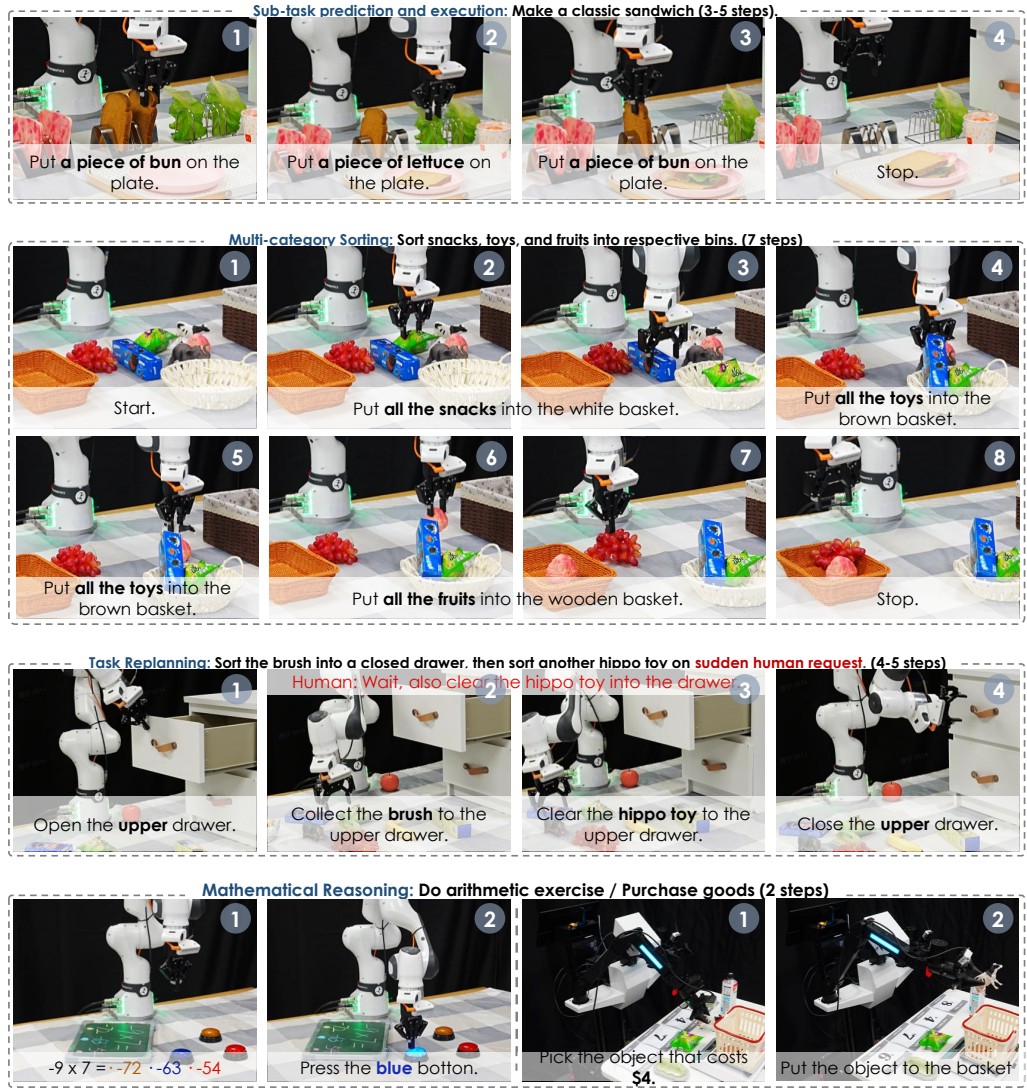

Figure 10: Showcases and results for long-horizon and reasoning manipulation.

# E CASE STUDY

To complement the quantitative results presented in previous sections, we provide qualitative case studies across simulation and real-world benchmarks to illustrate the versatility, robustness, and reasoning capabilities of ST4VLA in diverse manipulation scenarios. For video visualizations, please refer to the videos provided in the appendix and on our our website.

## E.1 CASE STUDY FOR PUBLIC BENCHMARKS

### E.1.1 CASE STUDY FOR SIMPLERENV

Figure 13 presents representative examples from the **SimplerEnv** benchmark, illustrating ST4VLA's performance across its canonical manipulation tasks. Each sequence shows the progression from initial scene perception and language instruction interpretation to successful task execution. These qualitative examples underscore the model's ability to ground natural language commands into actionable behaviors even in visually varied environments.

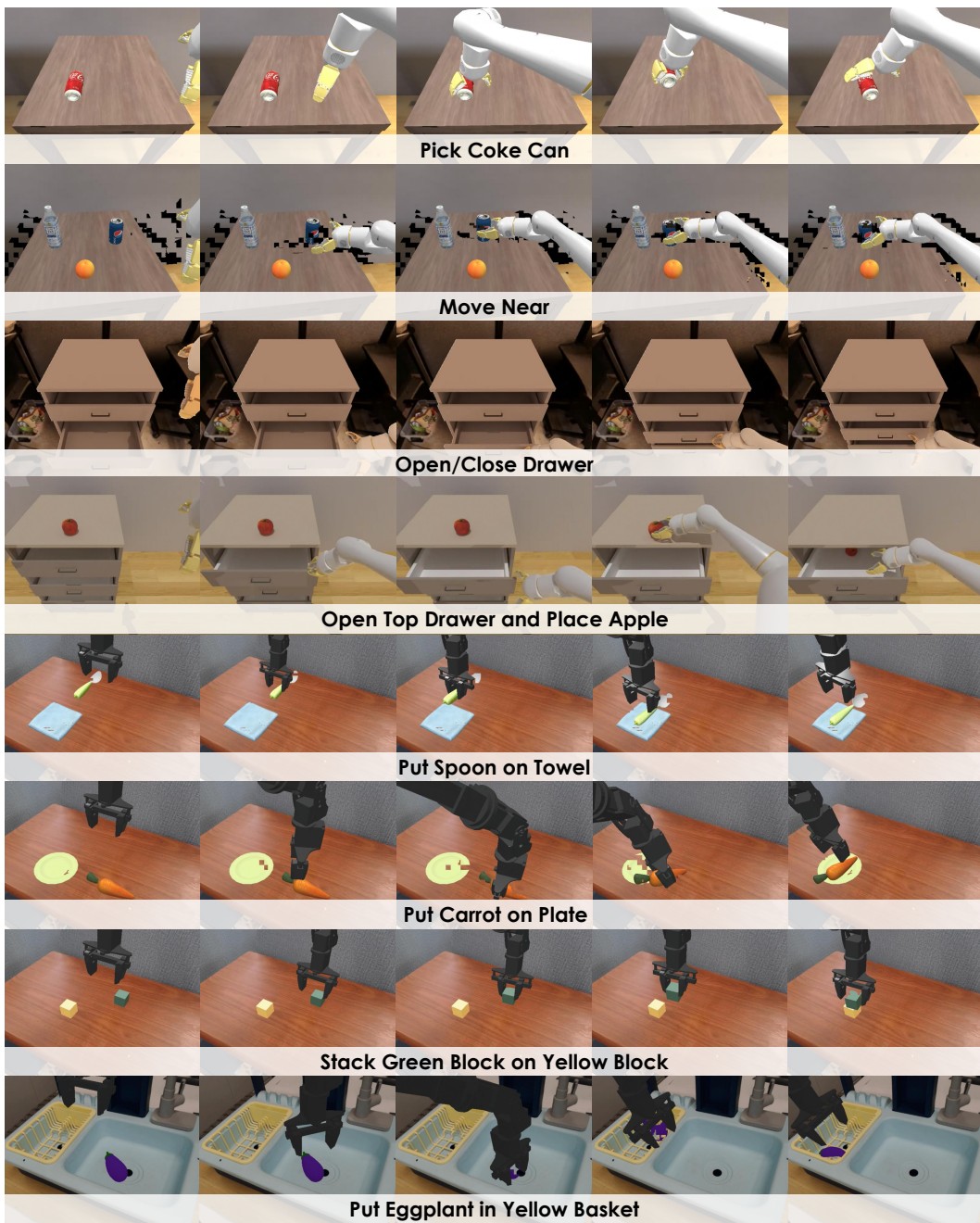

Figure 13: Evaluation showcases for SimplerEnv.

### E.1.2 CASE STUDY FOR LIBERO

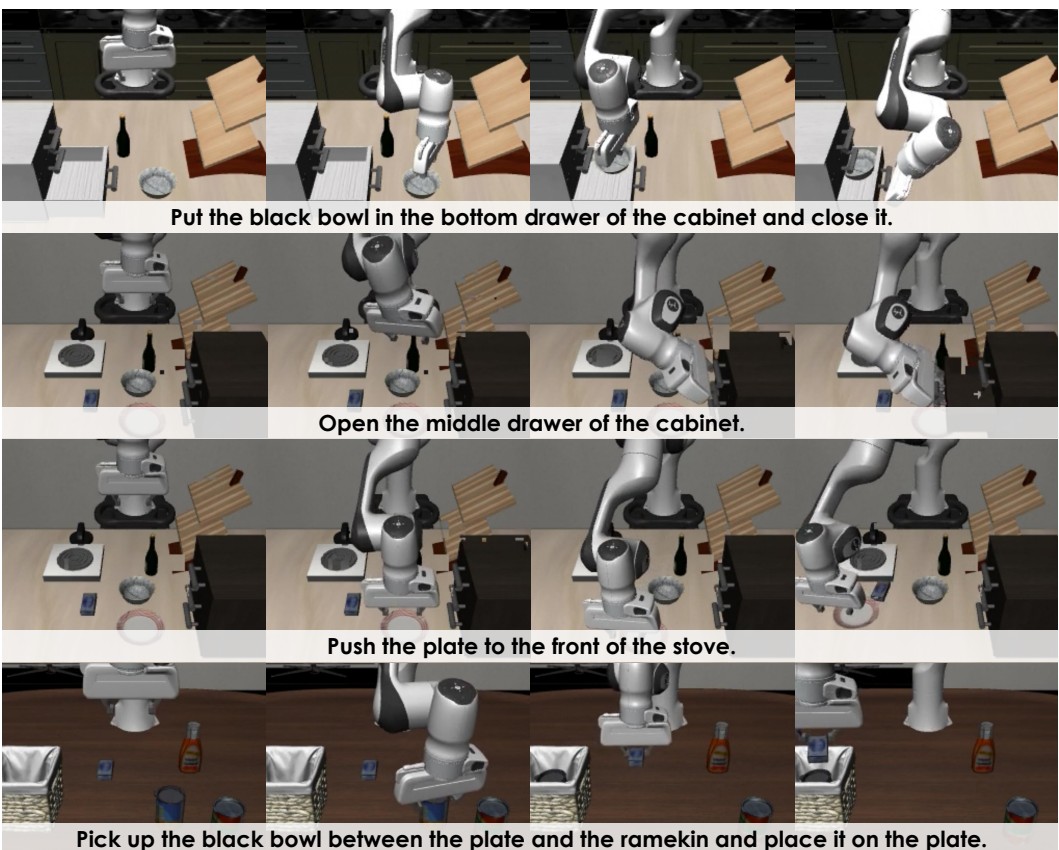

Figure 14: Evaluation showcases for LIBERO benchmark.

Figure 14 presents representative examples from the **LIBERO** benchmark. These cases collectively demonstrate ST4VLA's ability to interpret language instructions, adapt to spatial and semantic variations, and execute long-horizon plans with robust generalization.

## E.2 CASE STUDY FOR SIMULATED LARGE-SCALE PICK-AND-PLACE MANIPULATION

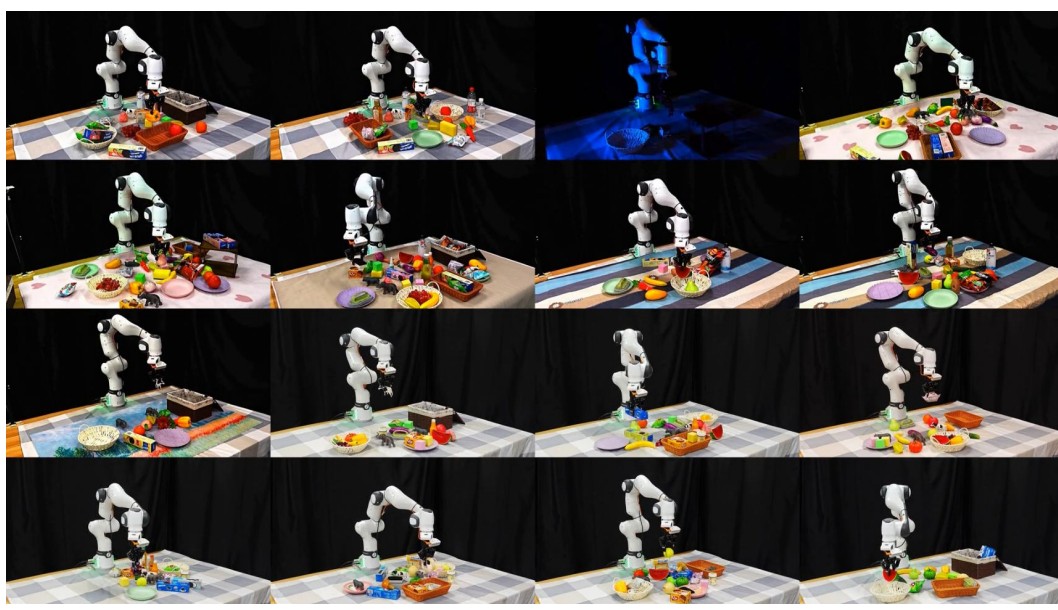

Figure 15: Showcases for simulated large-scale pick-and-place manipulation.

Beyond small-scale benchmarks, Figure 15 illustrates task executions in our large-scale simulated pick-and-place benchmark, which involves over 200 tasks and thousands of objects. These examples show that ST4VLA effectively parses instructions and manipulates previously unseen objects, while maintaining strong spatial grounding and generalization capabilities in cluttered, visually complex scenes.

## E.3 CASE STUDY FOR REAL-WORLD PICK-AND-PLACE MANIPULATION

Figure 16: Showcases for real-world large-scale pick-and-place manipulation.

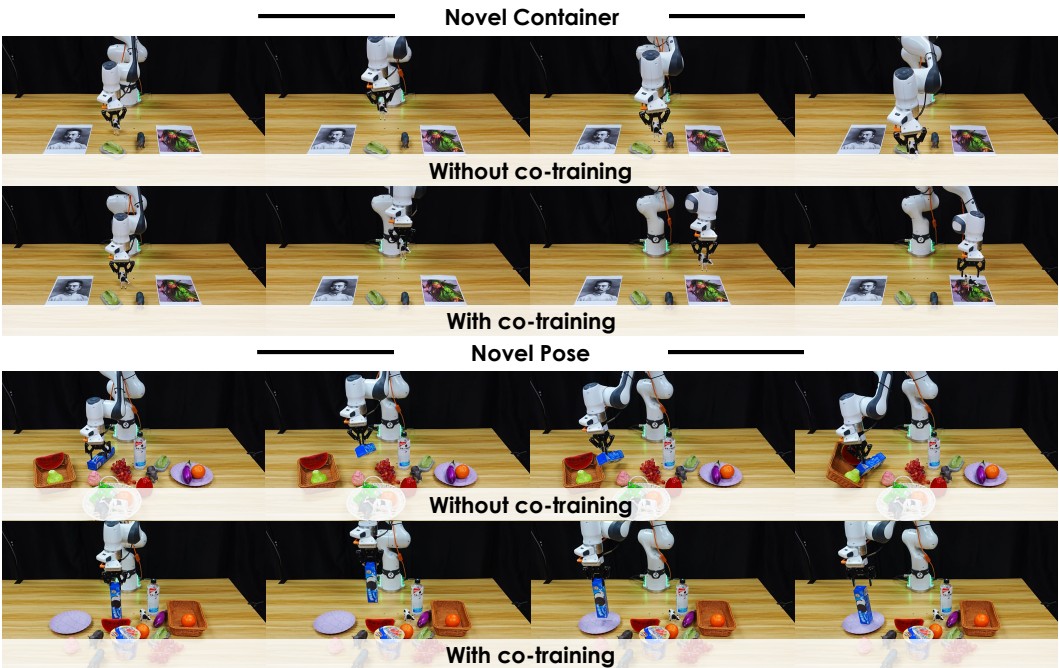

Figure 17: Showcases for real-world large scale pick-and-place manipulation w/wo co-training.

We further evaluate our framework in real-world cluttered tabletop environments. As shown in Figure 16, ST4VLA accurately interprets natural language instructions and completes single-step pick-and-place tasks involving numerous objects and containers under in-distribution, new background, unseen object, and unseen instruction conditions. Figure 17 further compares performance with and without co-training on VLM data, and shows that co-training substantially enhances robustness and generalization to novel objects and spatial configurations encountered during deployment.

### E.4 CASE STUDY FOR REAL-WORLD LONG-HORIZON MANIPULATION

We also evaluate ST4VLA on a range of long-horizon, reasoning-intensive tasks in real-world environments. Figures 18 to 22 collectively illustrate the model's reasoning and decision-making processes across diverse scenarios, including multi-step manipulation, numerical reasoning, object sorting, goods purchasing, and spatially constrained organization. These results demonstrate that ST4VLA effectively integrates perception, reasoning, and action planning, enabling robust execution of complex long-horizon tasks in dynamic real-world settings.

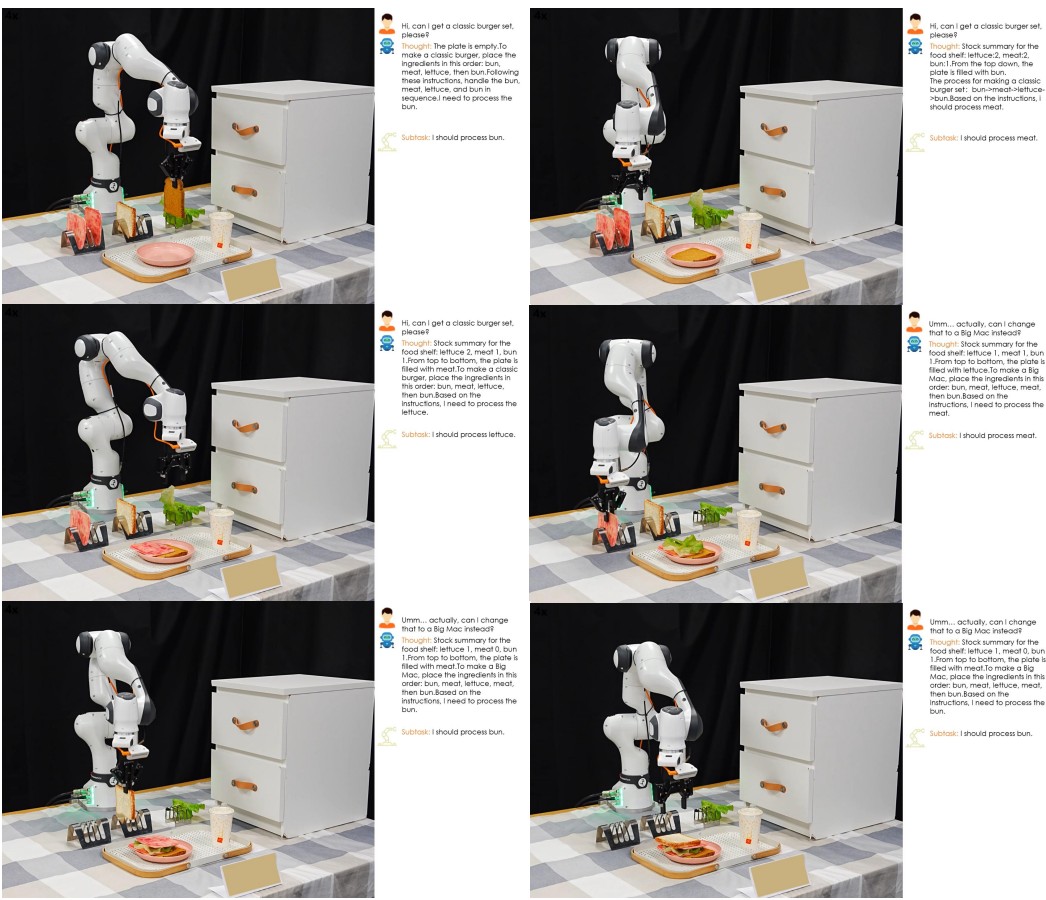

Figure 18: Showcases for making a sandwich.

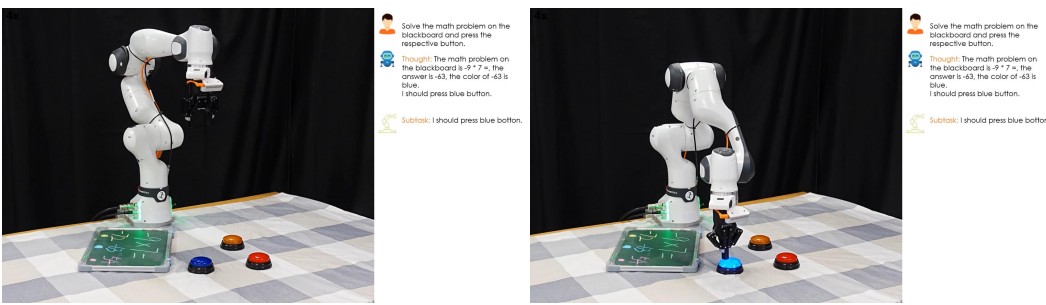

Figure 19: Showcases for math calculation.

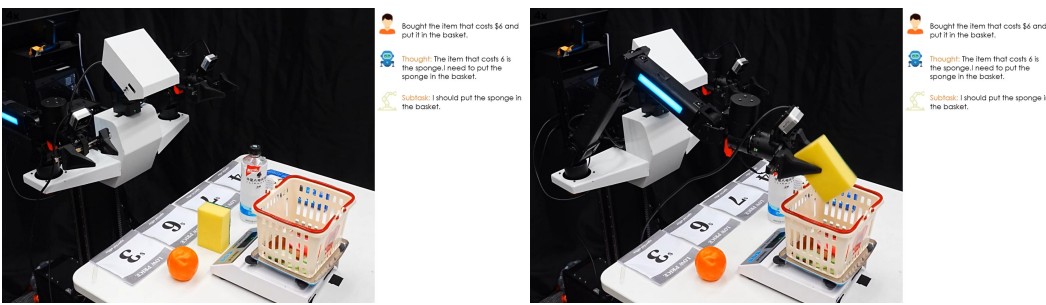

Figure 20: Showcases for sorting objects.

Figure 21: Showcases for purchasing goods.

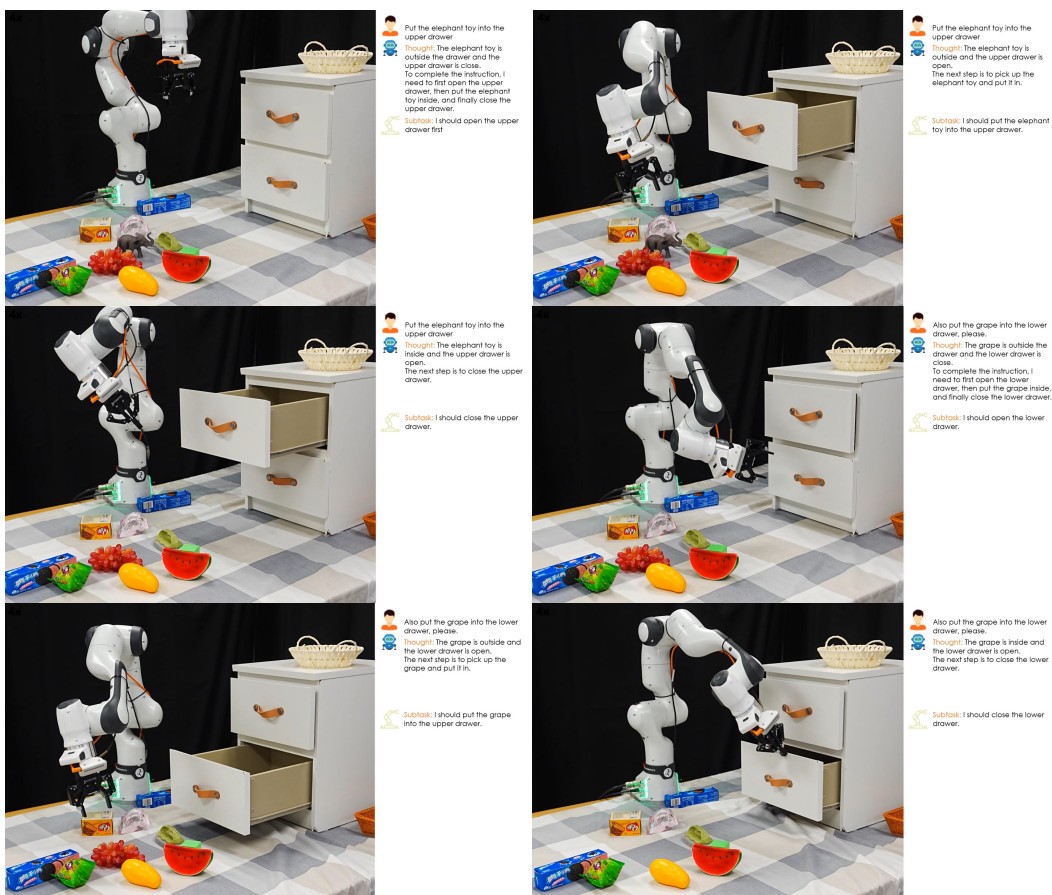

Figure 22: Showcases for sorting to drawer.

### E.5 FAILURE CASE STUDY

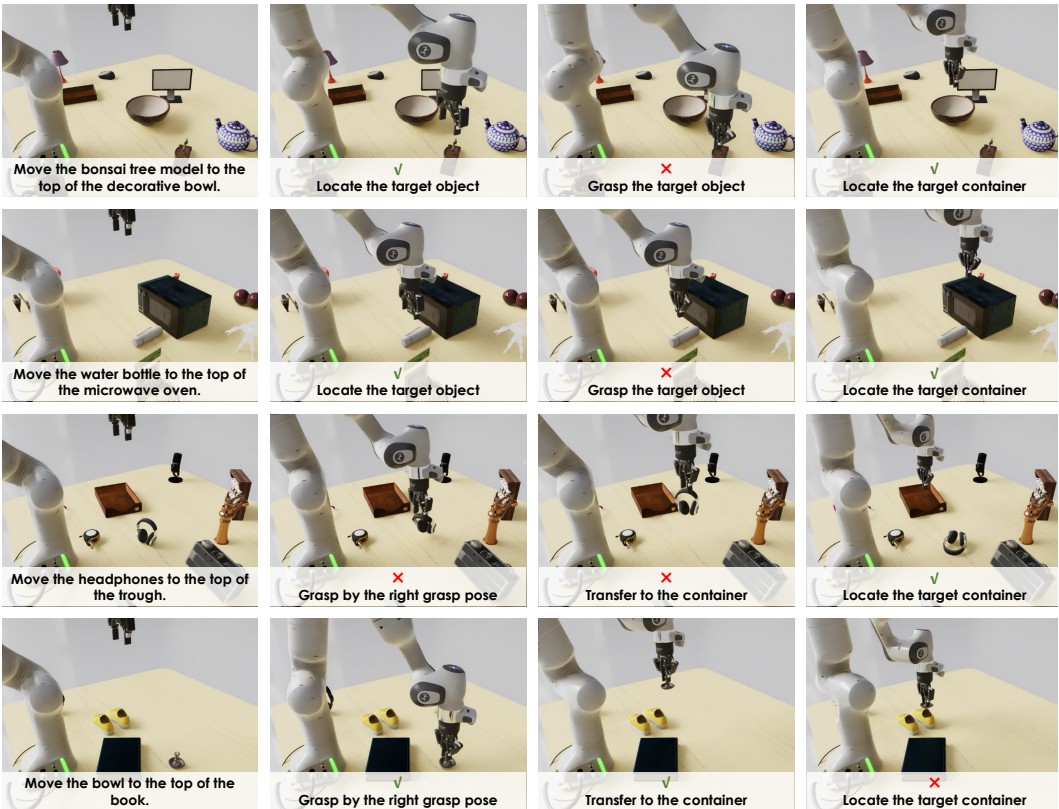

Figure 23: Failure case study.

To better understand the limitations of ST4VLA, we analyze representative failure cases during real-world instruction-following pick-and-place tasks. As shown in Figure 23. In some cases, the robot executes an incorrect grasp or misidentifies the target container, leading to task failure. These errors highlight challenges in robust perception and action grounding under cluttered environments. While some failures may stem from sensor limitations, integrating additional modalities, such as depth sensing and proprioceptive feedback, could improve performance. We leave this as future work to further enhance the reliability of instruction-conditioned manipulation in complex settings.

## F DATA

This section introduces the datasets used in ST4VLA, covering pre-training, mid-training, and post-training stages. For VLM pre-training, we construct large-scale spatial grounding datasets with point, box, and trajectory annotations to enhance spatial perception and vision-language alignment. Mid-training employs synthetic manipulation data to bridge pre-training knowledge and robotic execution. Post-training uses both simulated and real-world instruction-following data, including large-scale tabletop tasks and real-robot demonstrations for long-horizon manipulation.

### F.1 SPATIAL GROUNDING DATA FOR PRE-TRAINING

The multimodal training dataset for our model comprises over 3M data, categorized into four distinct types: General Question Answering (General QA), Bounding Box Question Answering (Box QA), Trajectory Question Answering (Trajectory QA), and Point Question Answering (Point QA), as shown in Figure 24 and detailed in Table 11. Notably, more than 2.3M of these data are dedicated to spatial reasoning datasets. These categories ensure robust multimodal understanding while supporting adaptation to embodied tasks in tabletop robotic scenarios. Below, we describe each category:

- **General QA**. Sourced from LLaVA-OneVision Li et al. (2024a) and InternVL3 Chen et al. (2024); Zhu et al. (2025), this category is sampled to cover diverse multimodal tasks, including image captioning, visual question answering (VQA), optical character recognition (OCR), knowledge grounding, and creative writing.
- **Bounding Box QA**. We curate a diverse collection of multimodal grounding datasets, including RefCOCO Yu et al. (2016); Mao et al. (2016), ASv2 Wang et al. (2024), and COCO-ReM Singh et al. (2024), sourced from InternVL3 Chen et al. (2024); Zhu et al. (2025). Additionally, we incorporate the ST4VLA Manipulation dataset, generated via scalable synthetic data generation as Appendix Section F.3, and the RoboRefIt dataset Lu et al. (2023), a specialized dataset for robotics grounding.
- **Trajectory QA**. This category integrates the A0 ManiSkill subset Xu et al. (2025a), the trajectory point subset from the ST4VLA Manipulation Dataset, and the MolmoAct dataset Lee et al. (2025) to enable precise end-effector trajectory prediction. The A0 ManiSkill subset provides high-quality, object-centric trajectory data, where small objects move in coordination with the robotic arm's gripper. These trajectories can be approximated as end-effector movements for tabletop manipulation tasks.
- **Point QA**. For precise point localization, we integrate multiple datasets, including the Pixmo-Points dataset Deitke et al. (2024), the RoboPoint dataset Yuan et al. (2024), the RefSpatial dataset Zhou et al. (2025a), and a point subset extracted from the ST4VLA Manipulation Dataset, each subjected to tailored preprocessing. Specifically, the Pixmo-Points dataset is filtered to exclude images with resolutions exceeding 1024 pixels and restricted to a maximum of 10 points per image. Additionally, we prioritize the extraction of object reference and region reference data from the RoboPoint and RefSpatial datasets to enhance grounding accuracy.

All point coordinates are converted to absolute coordinates to align with the Qwen2.5-VL Bai et al. (2025a) SmartResize prediction framework Bai et al. (2025b). Predicted coordinates are formatted in JSON and XML to support robust learning and adaptive processing of spatial instructions for diverse robotic tasks.

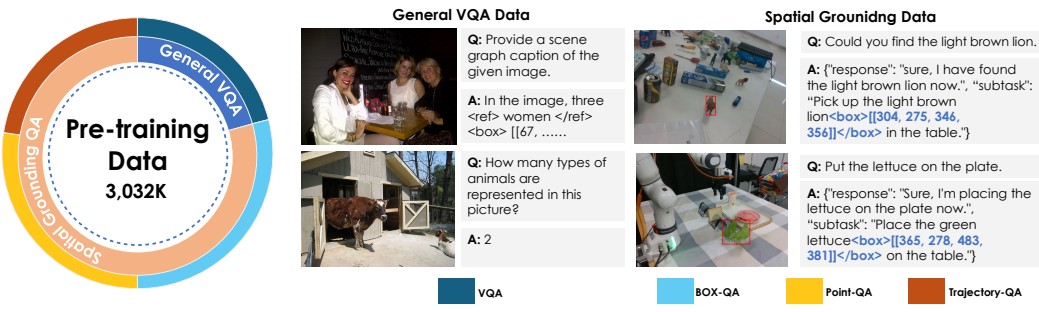

Figure 24: Overview of the pre-training data for the vision-language model. The data comprises two main parts: general VQA data to maintain the model's general multimodal capabilities, and spatial VQA data focusing on robotic-related grounding and spatial perception in a VQA format.

## F.2 Synthetic Data For Action Post-Pre-training

To bridge the gap between VLM and VLA, we introduce a Post-Pre-Training phase, where large-scale simulated data is used to pre-train the VLA after VLM pre-training. This stage initializes the action head and facilitates the learning of action representations. Post-Pre-Training requires maintaining diversity both at the instruction and object levels. We leverage GenManip as our data synthesis pipeline to construct a large-scale pick-and-place dataset which comprises 244K closed-loop samples. Specifically, we adopt the same object set and positional distributions as in InternData-M1 Data Contributors (2025), and process them through our scalable data pipeline. Each synthesized sample is rigorously validated to ensure correctness and consistency. To further enhance visual diversity, we introduce controlled randomization in lighting conditions and texture mappings.

Table 11: Pre-training multimodal datasets categorized by subtype (VQA, BBox-QA, Trajectory-QA, Point-QA), with scenario type and annotation method.

| Subtype | Dataset | Samples | Scenario | Annotation |
|---|---|---|---|---|
| VQA | AOKVQA | 33k | Web | manual |
| | ShareGPT4V | 182k | Web | automatic |
| | InternVL3 Proprietary Dataset | 225k | Web | automatic, manual |
| | COCOTextV2 | 16k | Web | manual |
| | VQAv2 | 82k | Web | manual |
| | TallyQA_COCO | 99k | Web | manual |
| BBox-QA | InternData-M1 (Bbox) | 431k | synthetic,real | automatic,manual |
| | RoboRefit | 36k | real | manual |
| | ASv2 | 128k | Web | manual |
| | COCO-ReM | 117k | Web | manual |
| | RefCOCO | 167k | Web | manual |
| Trajectory-QA | InternData-M1 (Traj) | 78K | real | manual |
| | A0 (ManiSkill) | 35k | synthetic | automatic |
| | MolmoAct (Traj) | 571k | synthetic | automatic |
| Point-QA | InternData-M1 (Point) | 114k | real | manual |
| | RefSpatial | 200k | synthetic | automatic |
| | RoboPoint | 422k | synthetic | automatic |
| | PixMo-Points | 96k | synthetic | manual |
| Total | – | 3.032 M | – | automatic, manual |

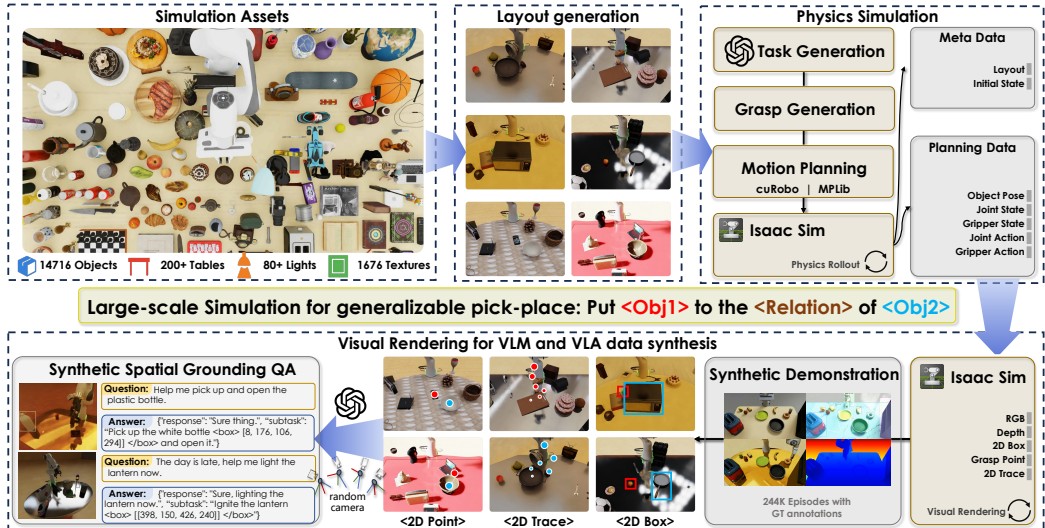

Figure 25: Simulation data synthesis pipeline. The pipeline generates diverse robotic manipulation data from a large asset library, converts intermediate representations into VQA data, and separates physics from rendering to reduce wasted failures and improve efficiency.

## F.3 SCALABLE SYNTHETIC DATA ENGINE FOR INSTRUCTION-FOLLOWING

To support large-scale end-to-end data generation for VLM pre-training, we build a highly scalable, flexible, and fully automated simulation pipeline on top of GenManip Gao et al. (2025) and Isaac Sim Makoviychuk et al. (2021).

**Automatic task synthesis for generalizable pick-and-place.** We develop a scalable simulation pipeline (shown in Figure 25) that generates diverse manipulation trajectories from randomized object layouts and lighting conditions. By leveraging privileged simulation signals, including object poses,

object meshes, and robot arm state, the system rapidly generates scene layouts via a scene graph solver and computes candidate grasps based on object meshes Liang et al. (2019). Each candidate trajectory is then executed once in physics for closed-loop verification, after which a scene-graph validator checks whether the task goals are achieved. Only trajectories that both execute successfully and pass validation are accepted, ensuring that all collected data are physically feasible and task-complete.

**Synthesis of VLM data and VLA data for Spatial Grounding.** For higher efficiency, robot planning and rendering are fully decoupled in our framework. The planner records structured scene and trajectory data, including joint states, object positions, and action information, which are later replayed by the renderer under randomized lighting, materials, and viewpoints. To align the simulation with the real world, we calibrate all cameras using ArUco markers, ensuring that their intrinsic and extrinsic parameters match those of real-world cameras, thus maintaining consistent viewpoint geometry. In addition to high-resolution images, the renderer produces rich intermediate outputs, such as object bounding boxes and 2D end-effector trajectories. These signals provide dense supervision for action learning and facilitate the creation of auxiliary datasets for tasks such as spatial grounding, affordance reasoning, and trajectory prediction. Our asset library includes 14K annotated objects, 211 tables, 1.6K textures, and 87 dome lights, offering data with high visual and physical diversity—critical for developing generalizable models.

## G  POST-PROCESSING OF TELEOPERATED DATA

### G.1  REAL TELEOPERATED DATA PROCESSED FOR EVALUATING LONG-HORIZON AND INTERACTIVE TASKS

**Data annotation.** To gather diverse tabletop task data, we placed a single-arm Franka robot on a lightweight mobile platform akin to DROID Khazatsky et al. (2024), enabling the robot to be easily transported and data to be captured across various scenes. We collected both short-horizon and long-horizon task data, with short-horizon tasks primarily involving pick-and-place operations, and long-horizon tasks including sandwich preparation, item sorting, and placing objects into drawers. For the long-horizon tasks, manual annotations were required to mark transitions between subtasks, and we preserved the action sequence annotations throughout the data collection process. After data collection, we segmented the data by marking specific time points. For example, the long-horizon task "make a classic sandwich" was decomposed into subtasks such as "place the bun on the plate," "put the meat in the sink," and "put the bun back on the plate." This decomposition allows the policy to first learn individual skill components in isolation before combining them into more complex behaviors.

**Construction of long-horizon tasks.** To enhance system efficiency and reduce the computational overhead caused by meaningless reasoning in the Visual Language Model (VLM), we adopt an agent-based approach to organize the VLM and VLA modules. VLM is triggered only when a human command is received or when the robotic arm stops moving. While VLA handles short-horizon tasks, VLM performs high-level semantic planning, enabling the system to support more complex long-horizon tasks.The data generation pipeline follows the structure outlined below.

**Keyframe extraction and data augmentation.** We begin by extracting keyframes from the collected data, including the first frame, last frame, and manually annotated task-switching frames. Given the limited amount of data, we extend the keyframes both forward and backward to maximize data utilization. For the forward extension, we identify moments of gripper changes near the keyframe and extract a portion (M%) of data from the segment between the gripper change moment and the keyframe, enhancing data diversity. Similarly, for the backward extension, we extract the next segment (N%) of data after the keyframe. For data balance, M is set to 60% and N to 40%, considering the data bias introduced by manual breakpoints.

**Scene layout extraction and action adjustment.** Once the data is augmented, we extract scene layout information, categorizing actions into three types: actions that have already occurred, actions that are yet to occur, and potential actions. For actions that are yet to happen, we adjust them by adding or removing actions. New actions are extracted from a potential action library, resulting in a new task list.

**Synthetic user instruction generation and response handling.** By combining this updated task list with the original instructions, we generate synthetic user instructions using GPT-4o-mini or a simple to-do list concatenation method. These instructions are further rewritten using GPT-4o-mini to handle vague or incorrect instructions and generate appropriate robot responses.

**Reasoning data generation and scene analysis.** Based on the actions that have occurred, actions that will occur, and the basic scene information, we generate reasoning data through templates or GPT-4o-mini. This reasoning data includes descriptions of the scene layout and action instruction analysis, which contributes to the large model's predictive content.

**Consolidation and formatting for model prediction.** Finally, we consolidate human instructions, robot responses, reasoning content, and next actions, and format them for the large model's predicted output. This ensures the system has a clear execution plan and can handle complex, long-horizon tasks.

