# OpenReview forum: "Spatially Guided Training for Vision-Language-Action Model"
_ICLR.cc/2026/Conference — ICLR 2026 Poster_

### Official Review · Reviewer_qHTz · 2025-10-25

**Soundness:** 3
**Presentation:** 2
**Contribution:** 2
**Rating:** 6
**Confidence:** 4

**Summary:**

This paper proposes SP-VLA, a spatially guided vision–language–action (VLA) model that introduces spatial grounding during both pretraining and post-training. By combining large-scale vision–language grounding data with robot-specific datasets, SP-VLA effectively aligns spatial and action objectives, leading to a robust and generalizable framework for robot learning that achieves significant improvements over vanilla VLA baselines on multiple benchmarks.

Overall, SP-VLA presents a well-motivated and promising approach to integrating spatial priors into general-purpose action representation learning. The reported gains are substantial, and the concept of latent spatial prompting is particularly compelling. However, the paper would benefit from clearer methodological exposition, more detailed attribution of performance gains, and improved consistency in presentation. With these refinements, it could make a strong contribution to the field.

**Strengths:**

1. SP-VLA achieves notable improvements over vanilla VLA across multiple settings.
2. The discussion around latent spatial prompting is particularly interesting. The authors recognize the difficulty of obtaining high-quality spatial annotations and instead design a lightweight strategy that implicitly guides visual representations. The qualitative motivation and empirical validation for this are compelling.
3. Stronger transfer to unseen objects and paraphrased language instructions; good long horizon reasoning.

**Weaknesses:**

1. Models such as GR00T and π0 are not cited when first introduced. Similarly, based on context (line 140-145), DiT appears to refer to a Diffusion Transformer but is never expanded upon. Clarifying these early and in figures would improve accessibility for readers new to the area.


2. The paper transitions quickly from the framework description to experiments without clearly specifying details like: (a) The training procedure and optimization details (loss functions, objective balancing). (b) Architectural modifications relative to existing VLAs, etc. These are crucial for reproducibility and clarity.

3. There is some ambiguity in experimental setups. For example: In Experiment 3.1, how are training losses weighted between spatial grounding and action data for the vanilla co-training baseline? In the RefCOCO-g perception task, what specific perception task is used, and how is vanilla VLA (without a dedicated perception head) evaluated on it?

4. SP-VLA includes multiple architectural components (spatial prompting, latent planner, diffusion-based actor, hierarchical reasoning). However, the paper attributes most gains to spatial grounding. It would strengthen the claims to discuss or experimentally isolate the contributions of each module. For example, how would performance change if the diffusion actor were replaced with a simpler transformer or MLP policy?

5. The paper does not clearly describe what forms of spatial prompts are used or whether they are consistent across tasks. Are these limited to the ones shown in Figure 2? Appendix D alludes to an ablation while making some conclusions about spatial prompts, but corresponding experiments seem missing. Given that spatial prompting is the key contribution, these things deserve a more detailed exposition (potentially in the main text).

6. Typographical and presentation issues:
* “WindowX” appears to be a typo for WidowX.
* Table 1 could improve readability by highlighting best results (e.g., bold/underline consistently to table 2).
* While the main text has a good number of visual explanation and diagrams, I think the ones related to spatial prompts, the ‘lightweight’ prompting module, etc. that center around the main contributions of the paper are missing.

**Questions:**

1. Why not directly fine-tune a pretrained VLM that already exhibits strong spatial grounding capabilities, instead of performing an additional spatial grounding pretraining stage? What are the concrete advantages of the proposed approach?

2. In Section 3.1, how are the spatial grounding and action losses weighted in the co-training baseline?

3. In Section 3.1, what specific perception task is used on RefCOCO-g, and how is vanilla VLA evaluated there without a perception head?

4. Could the authors clarify what kinds of spatial prompts are used (point, box, trajectory, etc.), and whether they vary across datasets or tasks?

5. How much do the diffusion transformer and hierarchical planner contribute independently to overall performance? Could similar benefits arise from these architectural components rather than spatial grounding alone?

6. Why do many popular baselines (Tables 2 and 3) underperform the “vanilla VLA” baseline used in this paper? What efforts are made to ensure that these baselines are evaluated under comparable settings?

---

> ### Author Response · Authors · 2025-11-26
> **Rebuttal to Reviewer qHTz [1/3]**
>
> We appreciate your recognition of SP-VLA’s significant performance gains over vanilla baselines, the compelling motivation and effectiveness of our latent spatial prompting strategy, and the model's robust generalization capabilities regarding unseen objects and long-horizon reasoning.
>
> ## W1: Related works cited and clarifying DiT.
> We appreciate your feedback regarding clarification of citations and acronyms. We have already incorporated these changes into the latest version of the manuscript to ensure improved clarity and proper attribution.
>
> ## W2: Training procedure and architectural details.
> (a) Training Procedure Details
> Our model is trained in two stages, transitioning from a standard Vision-Language Model (VLM) objective to a joint Vision-Language-Action (VLA) objective.
>
> Stage 1: Standard VLM Output and Loss (Qwen2.5-VL Pre-training)
> This stage follows the standard multimodal autoregressive objective, maintaining the VLM's powerful language generation capability. The model output is the multimodal autoregressive probability:
>
> $$
> p_{\theta}(y \mid x) = \prod_{t=1}^{T} p_{\theta}(y_t \mid x, y_{<t})
> $$
>
> where $x$ is the visual and text conditional input, which is encoded and fused into a token sequence by Qwen2.5-VL.
> The training objective is the Language Modeling (LM) loss:
>
> $$
> L_{LM}(\theta) = - E_{(x,y)\sim D} \left[ \sum_{t=1}^{T} \log p_{\theta}(y_t \mid x, y_{<t}) \right]
> $$
>
> Stage 2: Joint VLM and Action Output (VLA Post-training)
> In this stage, in addition to the Stage 1 Qwen2.5-VL text output, we introduce the Diffusion Transformer (DiT) for action output and incorporate a diffusion L2 loss. First, we obtain the conditional feature $z$ using the VLM + Q-Former + DINO:
>
> $$
> z = f_{\text{cond}}(x)
> $$
>
> We perform forward noise addition to the future action trajectory $a \in \mathbb{R}^{T\times d_a}$:
>
> $$
> x_t = q(a, t, \epsilon), \quad \epsilon \sim N(0, I), \quad t \sim U\\{0,\dots, T_{\text{diff}}-1\\}
> $$
>
> The DiT learns the noise predictor $\epsilon_{\phi}(x_t, t, z)$. The training objective is the Mean Squared Error (MSE) Action Loss:
>
> $$
> L_{act}(\phi) = E_{(x,a), t, \epsilon} \Big[ \big\|\epsilon_{\phi}(x_t, t, z) - \epsilon\big\|_2^2 \Big]
> $$
>
> The final loss for Stage 2 is the weighted sum of the Language Modeling loss and the Action loss:
>
> $$
> L(\theta,\phi) = L_{LM}(\theta) + \alpha \cdot L_{act}(\phi)
> $$
>
> where $\alpha = 10$.
>
> (b) Architectural Details
> Models such as $\pi 0$ utilize a layer-wise conditioned VLA (Vision-Language-Action) approach, meaning the hidden state from every layer of the VLM is used as a condition for the action head. In contrast, GR00T N1.5 uses only the intermediate hidden state of the VLM directly as the condition for the action head.
> Different from both of these approaches, SP-VLA uses a lightweight Q-Former to query the final hidden state of Qwen2.5-VL. This extracted feature is then combined with the DINO feature to serve as the condition for the action head. We optimize the action prediction using a diffusion denoise loss.
>
> ## W3 Ambiguity in experimental setups (Loss weighting & RefCOCO-g).
> We introduce a hyperparameter to balance the VLM and action losses. Specifically, the VLM's learning rate is set to $1 * 10^{-5}$, and the action head's learning rate is $1 * 10^{-4}$. The specific loss formulation is provided in the response to W2.
> For the RefCOCO-g task, we evaluate the bounding box (bbox) perception task. We follow the methodology of Qwen2.5-VL by having the model directly output the bbox in a text format, which eliminates the need for a dedicated perception head. Our Vanilla VLA (like many current mainstream VLAs) is constructed using a VLM + DP (Diffusion Policy) architecture and is trained exclusively on VLA data. Crucially, the VLM component retains its language head (which can predict the vocabulary), allowing it to directly output text. Furthermore, detailed settings for the Vanilla VLA are provided in Section 3.1.

---

> ### Author Response · Authors · 2025-11-26
> **Rebuttal to Reviewer qHTz [2/3]**
>
> ## W4 & Q5 How much do the Diffusion Transformer and Hierarchical Planner contribute?
> Thank you for the question. Yes, in Tables 2 and 3, our vanilla VLA is stronger than many prior baselines, and part of this gain can come from the **Diffusion Transformer (DiT) action expert** and the **hierarchical planner (VLM backbone)** choices.
>
> In our setup, the DiT action expert follows the CogACT design, while we replace the planner backbone with Qwen2.5-VL and the action expert. We provide a controlled comparison on WidowX (avg.):
>
> | Hierarchical planner | Action expert | Model        | WidowX avg. |
> |----------------------|---------------|--------------|-------------:|
> | PaliGemma            | DiT           | CogACT       | 51.3         |
> | Qwen2.5-VL           | DiT           | Vanilla VLA  | 54.7         |
> | Qwen2.5-VL           | Pi-FAST       | SP-VLA-FAST  | 58.6         |
> | Qwen2.5-VL           | MLP           | SP-VLA-MLP   | 44.2         |
> | Qwen2.5-VL           | π0            | SP-VLA-Pi-0  | 62.5         |
> | Qwen2.5-VL           | GR00T N1.5    | SP-VLA-GR00T | 63.6         |
> | Florence-2           | DiT           | SP-VLA-Florence  | 67.9     |
> | Qwen2.5-VL           | DiT           | **SP-VLA**   | **73.2**     |
>
> From **CogACT → Vanilla VLA**, changing the planner from PaliGemma to Qwen2.5-VL (with the same DiT expert) yields a **+3.7** gain. From **Qwen-OFT → Vanilla VLA**, switching the action expert from an MLP head to DiT yields a larger **+9.5** gain. Finally, **SP-VLA → Vanilla VLA** shows a **+18.5** improvement, indicating that our proposed SP-VLA training strategy contributes the most to achieving the final SOTA performance in this setting.
>
> Our primary comparison in the main paper is between **SP-VLA** and the **Vanilla VLA** baseline. In this setting, the only substantial difference is that SP-VLA introduces **spatial pre-training** and the **latent spatial prompting mechanism**. The sizable performance gap in this controlled comparison demonstrates that **spatial grounding provides substantial additional gains on top of an already strong diffusion-based policy**, rather than merely benefiting from a better policy head. This shows that spatial grounding is a key source of improvement **given a fixed, strong actor**.
>
> For clarity, what we referred to as a latent planner is more precisely a set of latent planning tokens: these are the final-layer VLM hidden states obtained under the spatial prompt.
>
> Taken together with the comparison to Vanilla VLA, these results indicate that **spatial grounding and the policy architecture are complementary**: spatial grounding delivers large gains when the actor is held fixed, and a strong diffusion actor is necessary to fully realize the benefits of the spatially grounded representation.
>
> ## W5: Spatial prompt forms, consistency across tasks, and missing ablations.
>
> Thank you for this insightful question. In our current implementation, we use a **single unified spatial prompt** across all tasks:
> > *“Figure out how to execute it, then locate the key object needed.”*
>
> Figure 2 shows three *optional* spatial **formats** (box / point / trace). Importantly, as described in Lines 192–199, **SP-VLA does not require the VLM to output bounding boxes / points / traces when post-training on action data**.
>
> That said, to answer the your concern about whether imposing explicit spatial-format requirements matters, we have run an ablation that varies the prompt format while keeping the training SimplerEnv protocol. We consider:
>
> - **Random padding:** `"xxx, xxx, xxx, xxx, xxx, xxx"`
> - **Unified prompting:** `"Figure out how to execute it, then locate the key object needed"`
> - **Box prompting:** Unified + `"Give the box coordinates according to the instruction"`
> - **Point prompting:** Unified + `"Your answer should be formatted as a list of tuples"`
> - **Trace prompting:** Unified + `"Based on the task description predict the trajectory that the end effector should take"`
>
> Results on SimplerEnv:
>
> | Prompt type | Google Robot VM | Google Robot VA | WidowX VM | Average |
> |---|---:|---:|---:|---:|
> | Random padding | 64.2 | 60.8 | 50.6 | 58.5 |
> | Unified prompting | **84.6** | **75.9** | 73.2 | **77.9** |
> | Box prompting | 80.9 | 73.0 | **75.8** | 76.6 |
> | Point prompting | 80.8 | 70.7 | 73.3 | 74.9 |
> | Trace prompting | 79.6 | 70.9 | 71.2 | 73.9 |
>
> These results show that (i) adding *any* meaningful spatial prompting strongly improves over random text, and (ii) the **unified prompt is already sufficient** and performs best on average, while explicit box/point/trace constraints do not consistently help. We have clarified this in Appendix D3 to make the spatial prompting setup and its consistency across tasks explicit.

---

> ### Author Response · Authors · 2025-11-26
> **Rebuttal to Reviewer qHTz [3/3]**
>
> ## W6: Typographical and presentation issues.
> We thank you for these helpful suggestions regarding the clarity and professionalism of the manuscript. We have already incorporated these corrections in the latest version of the manuscript.
>
> ## Q1: Why is the additional spatial grounding pre-training stage necessary?
> We confirm that we selected Qwen2.5-VL precisely because it exhibits the strongest spatial grounding capabilities among currently available open-source foundation models[1]. Although Qwen2.5-VL is powerful, its pre-training focuses on general-purpose scenes rather than specific ones. The key advantage of our proposed Stage 1 spatial grounding pre-training is to further elevate and specialize these capabilities for the robotics domain. General spatial grounding often lacks the fine-grained, high-precision reasoning required for physical manipulation tasks, especially concerning the position and interaction of the robot arm with objects. Our specialized Stage 1 bridges this gap, enabling SP-VLA to achieve robust and generalizable performance gains in robot-specific tasks.
>
> ## Q2: How are the losses weighted in the co-training baseline?
> Detailed information can be found in the response to W2.
> The ablation study can be found in Appendix D2.
>
> ## Q3: How is vanilla VLA evaluated on RefCOCO-g without a perception head?
>
> We use the standard RefCOCO-g referring expression grounding setup, where the model is given an image and a natural-language referring expression and is asked to output the target bounding box. Our Vanilla VLA is built in a **VLM + DP (policy)** manner and is trained only on VLA data; the underlying VLM still preserves its original VLM language header, so it can directly produce text outputs. The Vanilla configuration is described in Section 3.1. For evaluation, following the Qwen2.5-VL style, we instruct the model to output the bounding box in a **textual bbox format [x1,y1,x2,y2]**, so no separate perception head is needed.
>
> ## Q4: What types of Spatial Prompts are used?
> Stage 1 (Pre-training): We utilize three kinds of prompts, including box, point, and trajectory, to build robust spatial representations.
> Stage 2 (Post-training/Action): For all downstream robot manipulation tasks, we standardize the input by using only the box spatial prompt across all action datasets.
>
> ## Q5 How much do the Diffusion Transformer and Hierarchical Planner contribute?
>
> Please see W4.
>
> ## Q6: Why does our "Vanilla VLA" outperform many popular baselines?
>
> This is mainly because many popular baselines were developed in the OpenVLA codebase, which uses **PaliGemma** as the VLM backbone, while our implementation is built in our own codebase on a newer and stronger backbone (**Qwen2.5-VL**).
>
> To ensure comparable settings, all results in Tables 2 and 3 **strictly follow the SimplerEnv training protocol** and use **identical amounts of VLA data**. Methods that originally follow SimplerEnv’s data rules are run with the same number of demos, and all baselines are trained/evaluated under the same data budget and protocol.
>
> [1] Bai, S., Chen, K., Liu, X., Wang, J., Ge, W., Song, S., ... & Lin, J. (2025). Qwen2.5-vl technical report. arXiv preprint arXiv:2502.13923.

---

> > ### Comment · Reviewer_qHTz · 2025-11-28
> >
> > Thank you for the thorough rebuttal. I had many technical questions, and most have been adequately addressed. I appreciate the additional ablations, clarifications, typo fixes, and improved presentation; I think these additions help strengthen the paper’s contributions.
> >
> > I have one follow-up point. The reported +18.5 improvement for SP-VLA → Vanilla VLA is shown only with a DiT actor. It is difficult to conclude that the benefits of spatial grounding transfer similarly to other actor architectures (e.g., MLP) or to different planners/VLMs. Since spatially guided training is central to the paper, demonstrating that the method is architecture-agnostic would be important.

---

> ### Author Response · Authors · 2025-11-28
> **Response on Generalizability Across Architectures**
>
> Thank you for your positive feedback on our previous rebuttal. We greatly appreciate your recognition and would like to address your new insightful point regarding architectural generalizability.
> This concern aligns with questions raised by Reviewer QqBv about backbone dependence.
>
> To address this, we ran additional controlled experiments that vary both the planner/VLM backbone and the action expert. Across all settings, SP-VLA consistently improves over the corresponding Vanilla VLA baseline (e.g., 46.1 → 67.9 with Florence-2 as planner, and 30.7 → 44.2 with an MLP actor), supporting that our training recipe generalizes beyond a single architecture.
>
>
> For your convenience, we summarize the controlled results below:
>
> | Hierarchical Planner | Action Expert | Training Strategy | WidowX Avg. | Google VM Avg. | Google VA Avg. |
> | :--- | :--- | :--- | :---: | :---: | :---: |
> | Qwen2.5-VL | DiT | Vanilla VLA | 54.7 | 66.1 | 63.5 |
> | **Qwen2.5-VL** | **DiT** | **SP-VLA** | **73.2** | **84.6** | **75.9** |
> | Florence-2 | DiT | Vanilla VLA | 46.1 | 60.6 | 58.9 |
> | **Florence-2** | **DiT** | **SP-VLA** | **67.9** | **78.8** | **70.3** |
> | Qwen2.5-VL | MLP | Vanilla VLA | 30.7 | 50.5 | 51.3 |
> | **Qwen2.5-VL** | **MLP** | **SP-VLA** | **44.2** | **59.3** | **57.1** |
>
> [1] Xiao, Bin, et al. "Florence-2: Advancing a unified representation for a variety of vision tasks." Proceedings of the IEEE/CVF Conference on Computer Vision and Pattern Recognition. 2024.

---

### Official Review · Reviewer_BHsq · 2025-10-31

**Soundness:** 3
**Presentation:** 3
**Contribution:** 3
**Rating:** 6
**Confidence:** 4

**Summary:**

This paper proposes SP-VLA, a Spatially Guided Vision-Language-Action model that introduces latent spatial priors to bridge perception and action. The model conducts Spatial Grounding Pre-training on large-scale real and synthetic data to acquire spatial understanding, and then applies Spatially Guided Action Post-training to maintain such spatial priors during policy learning. It achieves new state-of-the-art results on SimplerEnv, and further validates the approach with a large-scale custom simulation benchmark.

**Strengths:**

- Achieves new SOTA performance on the SimplerEnv benchmark.
- Provides clear analysis of Figure 3, effectively demonstrating how spatial priors benefit action generation.
- Conducts extensive experiments on a large-scale, custom-built simulation to verify the generality of the approach.

**Weaknesses:**

- In Figure 3(b), Vanilla VLA’s curve may not have saturated. Given its long zero-phase and upward trend, it’s possible that Vanilla VLA could reach similar performance with longer training. Likewise, the Co-training setup doesn’t show a clear drop and might continue improving. I suggest extending training to 80–100 k steps or until convergence for all settings. It necessary to justify whether spatial priors mainly accelerate convergence or are essential for final performance.
- Moreover, it would be insightful to extend experiments of Figure 3(b) to Google Robot (VM/VA) setting
- Consider analyzing different scales of spatial prior pretraining data, not only training steps. More interpretability analysis on how spatial priors influence the latent representation would further strengthen the contribution.
- Related work currently misses several key papers that employ chain-of-thought style or multi-step reasoning in robot learning. Please cite:
1. Chain-of-Action: Trajectory Autoregressive Modeling for Robotic Manipulation
2. CoT-VLA: Visual Chain-of-Thought Reasoning for Vision-Language-Action Models
3. Efficient Robotic Policy Learning via Latent Space Backward Planning

**Questions:**

- How exactly is Figure 3(a) evaluated? Is the grounding metric still measured through spatial-ground prompts during Stage 2?
- Have you tried use co-training and your methods together.
- Why does the Spatially Guided Action Training curve in Figure 3(a) continue to improve, even though Stage 2 seemingly lacks explicit grounding supervision? This behavior seems counter-intuitive and deserves clarification.
- Do the authors plan to release the custom simulation benchmark used in large-scale experiments?

---

> ### Author Response · Authors · 2025-11-26
> **Rebuttal to Reviewer BHsq**
>
> We thank the reviewer for the positive assessment of our method's soundness and the extensive experimental validation. We appreciate the constructive suggestions regarding convergence analysis and data scaling, which have helped strengthen our submission significantly.
>
> ## W1, W2: Convergence Analysis & Extended Experiments
>
> Thank you for this rigorous suggestion. We have extended the training for all settings (including WidowX WM and Google Robot VM/VA) to 100k steps to verify convergence and performance boundaries. As shown in the **updated Appendix C.1** (line 1028), while baselines (Vanilla VLA and Co-training) eventually plateau, SP-VLA continues to improve and converges at a significantly higher performance level. This confirms that spatial priors do not merely accelerate convergence but are essential for achieving a higher performance upper bound that Vanilla VLA cannot reach even with prolonged training.
>
> ## W3: Effect of Spatial Prior Data Scale
>
> We investigated the scaling laws of spatial priors by varying the Spatial Grounding Pre-training data volume from 0M to 3M pairs, followed by Post-training on OXE. The results on SimplerEnv are as follows:
>
> | Pre-training Scale | Google Robot VM | Google Robot VA | WidowX VM | Average |
> | :--- | :--- | :--- | :--- | :--- |
> | 0 M | 66.1 | 63.5 | 54.7 | 61.4 |
> | 0.5 M | 66.1 | 61.2 | 55.6 | 61.0 |
> | 1.0 M | 68.9 | 65.5 | 55.8 | 63.4 |
> | 2.0 M | 72.8 | 72.9 | 67.3 | 71.0 |
> | 3.0 M | 84.6 | 75.9 | 73.2 | 77.9 |
>
> Our experimental results reveal a nonlinear relationship between spatial data scale and model performance. Performance gains remain modest when spatial grounding data is below 1.0M pairs. However, once the data scale surpasses 2.0M pairs, we observe dramatically increasing returns. At 3.0M pairs, the model achieves substantial improvement, with average performance rising from 61.4 to 77.9 - a remarkable 26.9% relative gain. These points have been updated in our revised version.
>
> ## W4: Missing Citations
>
> We agree that *Chain-of-Action*, *CoT-VLA*, and *Efficient Robotic Policy Learning* are excellent works on robotic reasoning. We have added a discussion of these papers to our Related Works section.
>
> ## Q1: How is Fig 3(a) evaluated? Is grounding still measured in Stage 2?
>
> Yes. During Stage 2 (Post-training), we continue to co-train with spatial-grounding data. The metric reported in Figure 3(a) is the Box IoU@0.5, measured on the validation set during this post-training phase.
>
> ## Q2: Have you tried using co-training and your methods together?
>
> Yes. Our method inherently integrates co-training. Multimodal data is utilized in both Stage 1 (Pre-training) and Stage 2 (Post-training) to ensure the alignment of spatial representations with action execution.
>
> ## Q3: Why does the Spatially Guided Action Training curve improve if Stage 2 seemingly lacks supervision?
>
> This was a misunderstanding caused by the visualization in our original Figure 2. In Stage 2, the VLM component continues to receive explicit supervision from Spatial Grounding data (via co-training). The spatial priors are actively maintained and refined. We have revised Figure 2 in the main paper to clearly illustrate this supervision flow.
>
> ## Q4: Release Simulation Benchmark and Assets
>
> We confirm that the custom simulation benchmark and the associated code will be open-sourced upon acceptance.

---

> > ### Comment · Reviewer_BHsq · 2025-11-28
> >
> > Thank you for the detailed and timely rebuttal. I appreciate the additional experiments and clarifications. I would like to raise two follow-up points:
> >
> > First, regarding the extended convergence analysis: since this experiment is critical to supporting the central claim that spatial priors raise the final performance ceiling rather than merely accelerate convergence, could the authors clarify whether they plan to release the training code and checkpoints associated with this analysis? The availability of these artifacts would greatly help the community verify the results.
> >
> > Second, since the authors mentioned releasing the large-scale simulation benchmark, I would also like to ask about the relationship between SP-VLA and the recent Intern-M1 / Intern-A1 works, which seem to share some similarity with this work. Could the authors clearly articulate the key differences and connections between SP-VLA and these Intern-series models and datasets?
> >
> > Overall, I appreciate the authors’ efforts in addressing the concerns and look forward to the final revision.

---

> ### Author Response · Authors · 2025-12-03
> **Open-Source Plan and Difference from the Intern-M1 / Intern-A1**
>
> Thank you for the insightful feedback and valuable suggestions.
>
> Regarding your first point, we will open-source the training code and model checkpoints associated with the extended convergence analysis to facilitate verification by the community and to further promote related research. The code and checkpoints are expected to be released alongside the final version of the paper.
>
> Regarding your second point, Intern-M1 / Intern-A1 technical report presents an earlier, broader, and more exploratory version of this research direction. It covers multiple aspects of the system and includes preliminary results.
>
> In contrast, the ICLR submission provides a focused and substantially extended version of the work, with several key improvements:
>
> - A refined and more mature method
> - New experiments and evaluations that were not included in the technical report
> - More rigorous analyses/ablations, such as the gradient consistency analysis in Section 3.1
> - Clearer problem formulation and improved theoretical and empirical justification

---

### Official Review · Reviewer_QqBv · 2025-11-01

**Soundness:** 3
**Presentation:** 2
**Contribution:** 2
**Rating:** 4
**Confidence:** 4

**Summary:**

The authors introduce a novel vision-language-action model, SP-VLA, that uses a two-stage training approach to ground action learning in spatial reasoning: 1) a pre-training stage, in which large-scale internet vision-language corpora and robot-specific datasets are formatted into a unified Question-Answer template and used to train a VLM, and 2) a post-training stage, in which a prompt that queries the spatial relationships between objects is appended to the task instruction and the VLM is trained to predict actions. The experiments evaluate SP-VLA on SimplerEnv, Libero, a new simulated benchmark based on IsaacSim, and short-horizon and long-horizon pick and place tasks in a real-world set-up. SP-VLA performs favorably compared to many VLAs on SimplerEnv, and is mainly compared to $\pi_{0}$ and GR00T N1.5 on the other benchmarks. The ablations show that SP-VLA relies on both training stages for its best performance.

**Strengths:**

- SP-VLA explicitly introduces a spatial grounding training phase and includes a loss over spatial metrics (box, point, trace).
- Strong results on multiple simulated and real-world benchmarks show that SP-VLA sets a new SOTA on SimplerEnv, Libero and the constructed benchmarks.

**Weaknesses:**

- Unclear whether the SOTA performance is due to the proposed training method or due to the VLM backbone. The comparisons between SP-VLA, $$pi_{0}$$ and GR00T N1.5 in Section 3 are fair if they have seen the same data during pre-training, but if that's not the case then it's possible the improvement comes from the strength of the VLM backbone, not necessarily the proposed training method.
- The details on the experiments and the losses are not very clear – see the questions below.

**Questions:**

l. 147-148 What are the outputs of the model in symbolic form? What is the loss during each stage?
l. 188 + l. 196 How do you get the spatial prompt? + Relatedly, can you explain the lightweight latent spatial prompting strategy?
l. 339 Are all the models in Tables 2 and 3 trained on the same amount/set of data?
l. 372 "benchmarks" <-- do you mean tasks?
l. 376 "post-pretraining" Is this different from the post-training phase, i.e. stage 2, of SP-VLA?
Regarding experiments in section 3: Did you finetune $$pi_{0}$$ and GR00T N1.5 on the tele-operation demonstrations before inference?
Table 4: The abbreviations are confusing, can you clarify them?
l. 1106 What prompt do you use to rephrase the instruction? Can you give an example of a rephrased prompt?
Section E.3: l. 1129 "unseen object position" Does that mean that the seen object is in a position unseen during training, or that an unseen object is placed in an unseen position? "unseen instruction" Does that mean it was rephrased or is it a novel task? l. 1161 Are positions of objects and containers fixed across trials or across models?
l. 1258 - 1260 How do the buttons correspond to the correct answer?
l. 1267 "be prompted" Is there a prompt from the user to tell the robot to continue to the next subtask? Also, are the subtask instructions generated by the model?

---

> ### Author Response · Authors · 2025-11-26
> **Rebuttal to Reviewer QqBv [1/2]**
>
> We sincerely thank you for the detailed and constructive feedback. We highly appreciate the positive recognition of our method’s soundness, the strong empirical results across simulated and real-world benchmarks, and the clear articulation of remaining concerns. Below we address the your primary questions and clarify methodological details.
>
> ## W1: Is the improvement mainly from the Qwen2.5-VL backbone rather than our proposed training method?
>
> We fully understand this insightful concern. To isolate the contribution of the training method from the backbone capacity, we provide three complementary pieces of evidence.
>
> - 1. We included a clear step-by-step ablation study in the original paper (Sec. 3.2)
>
>     Our paper includes carefully controlled comparisons against baselines that share the identical VLM backbone (Qwen2.5-VL-3B):
>
>     | Models                | Put Spoon on Towel | Put Carrot on Plate | Stack Green Block on Yellow Block | Put Eggplant in Yellow Basket | Avg  |
>     |-----------------------|---------------------|----------------------|-----------------------------------|--------------------------------|------|
>     | **GR00T N1.5**     | **75.3** | **54.3** | **57.0** | **61.3** | **61.9** |
>     | Vanilla VLA        | 56.6 | 63.3 | 27.0 | 71.8 | 54.7 |
>     | Vanilla Co-training VLA   | 70.3 | 68.4 | 20.5 | 85.2 | 61.1 |
>     | **SP-VLA**         | **80.2** | **79.2** | **35.4** | **98.0** | **73.2** |
>
>     The systematic improvement from Vanilla VLA (54.7) → Vanilla Co-training (61.1) → SP-VLA (73.2), under identical backbone conditions, directly attributes the performance gains to our proposed spatial grounding training stage rather than backbone capacity. Also, these results have been updated in our latest version.
>
> - 2. GR00T N1.5 already uses a significantly stronger VLM Backbone
>
>     It's important to note that GR00T N1.5 employs a significantly stronger VLM backbone than the Qwen2.5-VL-3B used in our SP-VLA. Public benchmarks demonstrate GR00T's VLM achieves superior object grounding performance:
>
>     | Model         | Size | GR-1 grounding IoU (↑) | RefCOCOg-val IoU (↑) |
>     |---------------|------|-------------------------|------------------------|
>     | Qwen2.5VL     | 3B   | 35.5                    | 85.2                   |
>     | GR00T N1.5 VLM | 2.1B | **40.4**                | **89.6**               |
>
>
>
> - 3. Rebuild SP-VLA with a weaker VLM backbone, i.g., Florence-2 [1]
>
>      We rebuilt SP-VLA using Florence-2 (a less capable model compared to Qwen2.5-VL and GR00T VLM) and observed the following results:
>
>     | Model | Put Spoon on Towel | Put Carrot on Plate | Stack Green Block on Yellow Block | Put Eggplant in Yellow Basket | Average Across Tasks |
>     |-------|-------------------|-------------------|---------------------------------|------------------------------|---------------------|
>     | GR00T N1.5 | 75.3 | 54.3 | 57.0 | 61.3 | 61.9 |
>     | Vanilla Co-training VLA | 75.2 | 31.3 | 3.1 | 75.0 | 46.1 |
>     | SP-VLA | 79.6 | 70.5 | 28.3 | 93.0 | 67.9 |
>
>     The results show that even with a weaker VLM backbone, SP training achieves performance comparable to GR00T (67.9 vs. 61.9), demonstrating the effectiveness of our training method, not only independent of the VLM's capacity.
>
>
>
> ## W2 & Q1 Clarifications on Outputs and Losses (l.147-148)
>
> 1. Stage 1: Spatial Grounding Pre-training
>
>     The VLM follows standard autoregressive decoding:
>     $p_\theta(y|x)=\prod_{t=1}^{T} p_\theta(y_t|x, y_{<t}), p_\theta(y_t|x, y_{<t}) = \mathrm{Softmax}(W_oh_t)$,
>
>     Loss: $\mathcal{L}*{\text{LM}} = -\sum\nolimits_t \log p \* \theta(y_t|x, y_{<t})$
>
> 2. Stage 2: Action Post-training
>
>     We condition DiT on VLM + Q-former + DINO embeddings: $z=f_{\text{cond}}(x)$.
>
>     Add forward diffusion noise: $x_t = q(a,t,\epsilon),\quad \epsilon\sim\mathcal{N}(0,I)$. Action diffusion loss: $\mathcal{L}\*{\text{act}} = |\epsilon\*\phi(x_t,t,z) - \epsilon|_2^2$. Final loss: $\mathcal{L} = \mathcal{L}\*{\text{LM}} + \mathcal{L}\*{\text{act}}$.
>
>
> [1] Xiao B, Wu H, Xu W, et al. Florence-2: Advancing a unified representation for a variety of vision tasks[C]//Proceedings of the IEEE/CVF Conference on Computer Vision and Pattern Recognition. 2024: 4818-4829.

---

> ### Author Response · Authors · 2025-11-26
> **Rebuttal to Reviewer QqBv [2/2]**
>
> ## Q2 Spatial Prompt Generation (l.188, 196)
>
> We clarify the process:
> - Base VLM with strong spatial grounding
>     - Qwen2.5-VL is chosen due to strong grounding ability.
> - Pre-training enhances grounding for robotic scenes
>     - Stage 1 significantly improves grounding robustness for real manipulation scenes.
> - Prompt selection
>     - We evaluate Stage-1 VLM using locate-related queries (e.g., “Where is the red object relative to the bowl?”) and identify prompts that reliably invoke spatial understanding.
> - Lightweight latent spatial prompting
>     During Stage 2:
>     - We do not require the VLM to output bounding boxes / points / traces.
>     - Instead, a lightweight Q-former extracts spatial features from latent tokens.
>     - These features condition the DiT action generator.
>
> This eliminates the need for heavy spatial annotations during Stage 2 and keeps post-training efficient.
>
> ## Q3: Data Fairness (l.339)
>
> All comparisons in Table 2 and Table 3 strictly follow the SimplerEnv training protocol and use identical amounts of VLA data. Works that originally follow SimplerEnv’s data rules directly use the same number of demos.
>
> ## Q4, Q5: Clarifications & Typos
>
> l.372: “benchmarks” → should be “tasks”. Updated in the new version.
> l.376: “post-pretraining” refers to large-scale training following Stage 2 paradigm. Will be clarified.
>
> ## Q6: Real-robot Settings & Table 4 (Finetuning, Abbreviations)
>
> Yes, all baselines are finetuned with the same tele-operation demonstrations.
>
> Table 4 abbreviations clarified in the revision:
> - In dist. = In-distribution
> - New inst. = New instance
> - Similar dist. = Similar distractors
> - New bg. = New background
> - Unseen obj. pos. / orient. = Position/orientation shifts
> - By attr = Attribute-based variation
>
> Full details in Appendix E.3.
>
> ## Q7: Prompt to Rephrase the Instruction
>
> ```python
> PROMPT_ATTR = """
> The task is to optimize an existing instruction for a robot model.
>
> ### Input:
> - **pick obj description**: {Object description obtained from simulation, e.g., red apple}
> - **container description**: {Container description obtained from simulation}
> - **Raw Instruction**: {Original instruction: Move obj1 to the top of container}
> Rewrite guideline: Focus on {rewrite_type:materials/color/shape}
>
> ### Rules:
> 1. There are many items on the desktop. Ensure the rewritten instruction is specific enough to unambiguously identify the target object and container.
> 2. If the attributes mentioned in the original instruction (such as shape, color, or material) are not sufficient to uniquely identify the object, add extra features (e.g., relative position, size, or additional visual properties) to remove ambiguity.
> 3. Do NOT mention the object’s common name (e.g., do not say "apple", "cup", etc.).
> 4. The rewritten instruction must sound natural and fluent while preserving the original meaning.
>
> ### Output:
> - Provide **5 examples** of optimized instructions in JSON format (a list of strings).
> - Keep all examples simple, clear, and easy to understand.
>
> ### Example:
> input: Place the apple to the top of plate.
> output: [
>     "Put the red sphere on top of the round white plate.",
>     "Stack the small red object onto the large white plate.",
>     "Set the red fruit on the circular plate.",
>     "Position the shiny red sphere on top of the white ceramic plate.",
>     "Move the red object closest to the center onto the round plate."
> ]
> """
> ```
>
> ## Q8, Q9, Q10, Q11: Additional Clarifications from Appendix (E.3 and others)
>
> - “Unseen object position”: seen objects placed in novel regions not seen during training.
> - “Unseen instruction”: same task, but rewritten by GPT to test language robustness.
> - Object positions are fixed across trials and restored across all models.
> - Button–answer mapping: each answer on the blackboard is written in a distinct color, and the physical buttons use the same set of colors; the correct answer is indicated by matching the color of the number on the blackboard with the corresponding button.
> - Long-horizon tasks:
>     - No human prompting required between subtasks.
>     - The VLM Planner autonomously outputs the next subtask instruction.
>     - All instructions are generated by the same VLM.

---

### Official Review · Reviewer_uJaW · 2025-11-01

**Soundness:** 3
**Presentation:** 3
**Contribution:** 2
**Rating:** 4
**Confidence:** 4

**Summary:**

The paper proposs SP-VLA, using spatial prior to bridge the language space and action space. The SP-VLA has two stages, the first train the VLM with image, language, and spatial priors like bbox, points or trajectories. The second stage use the latent generated in the first stage to train a DiT policy for control.

**Strengths:**

1. The motivation of the paper is good, which using spatial prior to pretrain the vlm, then further tune the vlm for downstream tasks, which compare with mainstream vlm which focus on daily question or specific QA task, spatial information seems more important for control and robotic task.

2. The paper deliver thourough experiments and ablations to claim the spatial priors both in pretraining and post-training will help the embodied control.

**Weaknesses:**

1. The idea of use spatial priors to pretrain or post train or cotrain a vlm is not novel and actually used by many recent papers like , so i do not think the proposed spatial prior will help both pretraining and post training is novel enough.

2. In the experiment design, the author try to verify with spatial prior in pretraining or post training, the SP-VLA can perform better than other fundational VLAs, but it seems the compared method like pi and openvla, these paper does not claim they use specific priors, The author might need to compare with more related work as mentioned in 1.


[1] Wen, Chuan, et al. "Any-point trajectory modeling for policy learning."
[2] Niu, D., Sharma, Y., Biamby, G., Quenum, J., Bai, Y., Shi, B., ... & Herzig, R. (2024). Llarva: Vision-action instruction tuning enhances robot learning.

**Questions:**

See weakness part, and furthermore, small questions:
1. for the real world experiment, section 3.4, it seems all task is pick and place. It is not clear clarified what is the experiment setting? Did the author use standard DROID? If so, why the pi-zero is so bad on unseen object, with my own experience, at least pi-zero should have better results than what record in Table 4.

---

> ### Author Response · Authors · 2025-11-26
> **Rebuttal to Reviewer uJaW [1/2]**
>
> Thanks for your insightful feedback and acknowledgement of our comprehensive experiments and well-motivated ideas.
>
> ## W1: The idea of using spatial priors to pretrain or post post-train or cotrain a VLM is not novel.
>
> We agree with you that spatial priors are not new concepts. The contribution of our work does not lie in introducing spatial cues themselves, but in addressing the open question of how to scale spatial priors effectively in end-to-end VLA training so that they yield reliable and significant gains.
>
> Similar to how the GPT series demonstrated the impact of effective scaling strategies rather than architectural novelty, our work focuses on identifying a practical and scalable recipe for leveraging spatial priors. Existing VLAs incorporate spatial cues, but they do not show consistent improvements under large-scale instruction-following control experiments in both simulation (3000+ objects) and real-world clustered scenes (with more than 20 objects), long-horizon tasks (over 10 subtasks).
>
> SP-VLA demonstrates the full-stack spatial guided action training, from large-scale data curation (built on top of over 1.4W 3D assets for scalable simulation, leading to over 3M+ cross-interface spatial priors (e.g., point, box, traces)), spatially guided training, adequate experiments on both large-scale and reproducible simulation, and real-world clustered or long-horizon instruction-following tasks. We hope these findings offer actionable and scalable insights for scaling spatial structure in modern VLA systems.
>
> Finally, we clarify the technical difference with [1,2] here: Any-point trajectory modeling [1] focuses on predicting arbitrary point tracks and then using the predicted traces as an intermediate input for policy learning, rather than a unified prompt-based interface that supports multiple spatial forms and directly guides action training. LLaVRA [2] uses 2D trajectories as spatial priors and represents actions in text; we cite it as an attempt to integrate spatial priors into VLA models. Compared with LLaVRA, SP-VLA builds on this direction with a VLM backbone and a DiT-based policy head, and expands spatial priors beyond trajectories to include boxes, points, and traces.  Yet, these works are insightful, and we include them in our related works section.
>
> ## W2: Lack experimental comparison with spatial-prior VLA works.
>
> Thanks for the suggestion to compare with more spatial-prior VLA methods. We agree this is important, but direct one-to-one reproduction is often not feasible: some recent systems are not fully released (for example, MolmoAct), and many VLA papers use different datasets, robot embodiments, observation/action spaces, and model backbones, which makes “drop-in” comparisons hard and sometimes not meaningful.
>
> That said, we add direct comparisons on shared benchmarks where strong spatial-aware VLAs report results. On SimplerEnv, to the best of our knowledge, SP-VLA achieves the best reported performance among existing VLAs. The compared models include SpatialVLA, which explicitly augments VLA inputs with robotics-specific 3D-aware signals, and Magma / GR00T N1.5, whose spatial capabilities largely come from large-scale pretraining.
>
> | Models | Google Robot VM | Google Robot VA | WidowX VM | Average |
> |---|---:|---:|---:|---:|
> | Magma | 52.9 | 51.6 | 35.8 | 46.8 |
> | GR00T N1.5 | 35.2 | 44.5 | 61.9 | 47.2 |
> | SpatialVLA | 75.1 | 70.7 | 42.7 | 62.8 |
> | SP-VLA | 84.6 | 75.9 | 73.2 | 77.9 |
>
>
> In addition, we compare to work Any-point trajectory modeling for policy learning [1] on LIBERO (Franka), where results are reported on the same benchmark split. SP-VLA consistently outperforms prior spatial-aware baselines across all task groups:
>
> **Table 5: Result comparisons of robotic manipulation on LIBERO (Franka) benchmark.**
>
> | Models | Spatial | Objects | Goal | Long | Avg |
> |---|---:|---:|---:|---:|---:|
> | ATM Diffusion Policy [1]  | 71.0 | 89.0 | 58.7 | 44.0 | 62.9 |
> | SpatialVLA | 88.2 | 89.9 | 78.6 | 55.5 | 78.1 |
> | GR00T N1.5  | 92.0 | 86.0 | 92.0 | 76.0 | 86.5 |
> | **SP-VLA** | **98.0** | **99.0** | **93.8** | **92.6** | **95.9** |
>
> ## Q1-1: Real-World Setup Clarification (Sec. 3.4)
> We did not use the standard DROID setup. Instead, we utilized a Franka Emika Panda robot with a specific camera configuration (one wrist-mounted, one third-person view) designed for the field of view for our instruction-following tasks, as illustrated in Appendix E.5 (Figure 11). Due to space constraints, the detailed experimental setup, including evaluation metrics, is provided in Appendix E.3.

---

> ### Author Response · Authors · 2025-11-26
> **Rebuttal to Reviewer uJaW [2/2]**
>
> ## Q1-2: Why $\pi_0$ performs poorly in Sec. 3.4 (Table 4)
>
>
> Thank you for pointing this out. You noted that $\pi_0$ typically performs well on standard tasks, and we agree that $\pi_0$ is a strong baseline. Its lower performance in Table 4 mainly comes from the specific difficulty of our **Cluttered Environment** setting: we place **10–20 distractor objects** on the table to strictly stress **instruction following** and **target localization** under heavy clutter.
>
> We attribute $\pi_0$'s performance drop in this setting to two main factors:
>
> 1. Data Distribution Shift: While DROID data is diverse, it does not densely cover scenarios with such a high density of clutter (10–20 distractors) combined with complex language instructions. Importantly, this behavior is consistent with the concurrent work GR-3 [3], which explicitly evaluates $\pi_0$ under similarly cluttered real-world setups and reports that $\pi_0$ also degrades substantially (as listed in Figure 7(a)). This external evidence suggests that our protocol is not an outlier, but a realistic stress test for instruction grounding under heavy clutter.
>
> 2. Lack of Multimodal Co-training: Our hypothesis is supported by findings from $\pi_{0.5}$, which show that removing web-scale multimodal data (image-text, VQA, grounding data) significantly degrades performance on OOD object localization and instruction following. $\pi_0$, lacking this extensive multimodal co-training, struggles to ground open-vocabulary instructions amidst heavy clutter, effectively losing "open-world generalization" capabilities in this specific regime.
>
> ----
> [3] Cheang, C., Chen, S., Cui, Z., Hu, Y., Huang, L., Kong, T., ... & Yang, Y. (2025). Gr-3 technical report. arXiv preprint arXiv:2507.15493.

---

### Official Review · Reviewer_MEHz · 2025-11-01

**Soundness:** 3
**Presentation:** 3
**Contribution:** 3
**Rating:** 6
**Confidence:** 3

**Summary:**

This paper presents SP-VLA, a dual-system VLA framework that integrates action learning with spatial priors. The framework consists of two components: Sys2, a VLM (Qwen2.5-vl-3B), and Sys1, a DiT-based diffusion model. A Q-former module links the two systems through cross-attention, compressing Sys2’s tokens into a fixed-length representation that is then injected into Sys1.

Training occurs in two stages: spatial grounding, where the VLM acquires spatial priors (points, boxes, and trajectories) from a large dataset, and spatially guided action post-training. Experimental results in both simulated and real-world environments demonstrate that SP-VLA outperforms existing baselines.

**Strengths:**

The paper is well-organized and clearly written, making it easy to follow. The authors provide extensive implementation details to support reproducibility and include numerous illustrative examples that enhance understanding. The experiments are reasonable and solid.

**Weaknesses:**

The introduction of spatial priors to enhance VLA performance is conceptually intuitive and has been explored extensively in prior research (e.g., LLaRA [1], TraceVLA [2], Magma [3], among others). Similar strategies, such as dataset curation via object detection, have also been commonly adopted. Moreover, the dual-system design has been previously discussed in works like TinyVLA [4] and $\pi_0$ [5]. Consequently, the paper’s contribution appears to offer limited novelty relative to existing literature.

Nonetheless, the work provides substantial implementation details and practical engineering insights, which could be valuable to the research community. Therefore, I recommend a weak acceptance.

Reference

1. Li, X., Mata, C., Park, J., Kahatapitiya, K., Jang, Y. S., Shang, J., ... & Ryoo, M. S. (2024). Llara: Supercharging robot learning data for vision-language policy. arXiv preprint arXiv:2406.20095.

2. Zheng, R., Liang, Y., Huang, S., Gao, J., Daumé III, H., Kolobov, A., ... & Yang, J. (2024). Tracevla: Visual trace prompting enhances spatial-temporal awareness for generalist robotic policies. arXiv preprint arXiv:2412.10345.

3. Yang, J., Tan, R., Wu, Q., Zheng, R., Peng, B., Liang, Y., ... & Gao, J. (2025). Magma: A foundation model for multimodal ai agents. In Proceedings of the Computer Vision and Pattern Recognition Conference (pp. 14203-14214).

4. Wen, J., Zhu, Y., Li, J., Zhu, M., Tang, Z., Wu, K., ... & Tang, J. (2025). Tinyvla: Towards fast, data-efficient vision-language-action models for robotic manipulation. IEEE Robotics and Automation Letters.

5. Black, K., Brown, N., Driess, D., Esmail, A., Equi, M., Finn, C., ... & Zhilinsky, U. (2024). $\pi_0 $: A Vision-Language-Action Flow Model for General Robot Control. arXiv preprint arXiv:2410.24164.

**Questions:**

N/A

---

> ### Author Response · Authors · 2025-11-26
> **Rebuttal to Reviewer MEHz [1/2]**
>
> We appreciate your recognition of our solid experiments and engineering insights. Below we address the novelty concerns in details.
>
> ## Main concerns: novelty concerns of spatial priors.
>
> We agree with you that spatial priors are not new concepts. The contribution of our work does not lie in introducing spatial cues themselves, but in addressing the open question of **how to scale spatial priors effectively in end-to-end VLA training so that they yield reliable and consistent gains**.
>
> Analogous to how the GPT series demonstrated that scaling strategies—not architectural novelty—drive major advances, our work identifies a practical, scalable recipe for integrating spatial priors into modern VLA systems. While prior VLAs incorporate spatial signals, they do not demonstrate consistent improvements under controlled, large-scale instruction-following evaluations in both simulation (3,000+ objects) and real-world environments (clustered scenes, long-horizon tasks).
>
> SP-VLA closes this gap by providing a full-stack spatial-guided action learning pipeline: (i) large-scale spatial-prior curation derived from 14K+ 3D assets and over 3M cross-interface spatial annotations (points, boxes, traces), (ii) spatially guided pretraining and action learning, and (iii) extensive experiments that show reproducible gains across synthetic and real-robot settings. We hope these findings offer actionable insights for scaling spatial structure in VLA systems.
>
> ## W1: Similar related works of spatial priors methods.
>
> - LLaRA is not end-to-end: it predicts pixel-level coordinates and action primitives that are decoded by a downstream rule-based controller. SP-VLA is fully data-driven and end-to-end, with no rule-based parsing.
>
> - TraceVLA uses visual traces only as history encoding. It cannot predict future spatial traces. SP-VLA pretrains on large-scale synthetic/web data to predict future visual traces, providing explicit spatial guidance for action planning.
>
> - Magma is architecturally closer, but it relies on direct action post-training. In contrast, SP-VLA introduces ``spatial prompting'', which explicitly activates pretrained VLMs’ spatial-planning ability to guide action learning. As shown in Fig. 3(b), this yields ~2.5× faster convergence and stronger final performance.
>
> - Most closest to ours is concurrent work $\pi_{0.5}$, which jointly trains multimodal (grounding data) and action data, but relies on 8+ months of real-robot data and does not evaluate large-scale object-level instruction following. Differently, SP-VLA constructs a large, cross-interface synthetic spatial-prior dataset and demonstrates that spatial guided training yields more stable action improvements than direct co-training.
>
> **Key empirical findings**. Both Magma and $\pi_{0.5}$ perform vanilla co-training. In our fair comparison, however, as shown in Fig. 3(b) and Fig. 6, vanilla co-training of spatial prior and action data does not reliably improve action performance. Our gradient analysis (Fig. 3(c)) shows low alignment between spatial grounding and action prediction gradients. SP-VLA increases this alignment, enabling spatial priors to meaningfully guide action learning—something that Magma and $\pi_{0.5}$ do not address.
>
> However, these works are insightful, and we include them in our related works section.
>
> ## W2: Similar related works on dataset curation via object detection.
>
> We appreciate the connection. While dataset curation is a shared goal, our simulation pipeline provides several beneficial points beyond real-world ones:
> - Prior works (e.g., LLaRA, Magma) rely on off-the-shelf detectors to produce pseudo-labels, which can be inconsistent under occlusion. SP-VLA employs simulation (Isaac Sim) to obtain pixel-perfect ground-truth 2D masks and bounding boxes, eliminating detector noise.
> - Prior works rely on real data with a limited set of physical objects. Our simulation leverages large-scale 3D assets (14K+ objects), 200+ tabletop configurations, and 1,676 textures to synthesize diverse and scalable robot demonstrations.
>
> Thus, although it shares the high-level goal of large-scale data curation, SP-VLA uniquely leverages diverse assets and robot synthesis to produce reliable spatial priors that demonstrably improve downstream action learning.

---

> ### Author Response · Authors · 2025-11-27
> **Rebuttal to Reviewer MEHz [2/2]**
>
> ## W3: Similar related works on dual-system (TinyVLA [4] and pi0.5)
>
> We appreciate the connection. We emphasize that the dual-system architecture itself is not the main contribution of our work. Our primary contributions are the spatially guided training framework that explicitly aligns action optimization with spatial grounding.
>
> That said, our design makes a practical and empirically validated choice: we connect System 2 VLM and System 1 action expert through a query-based Q-Former, which compresses diverse input tokens into a stable fixed-length representation. This differs from prior dual-system VLAs and provides improved stability (+6.8 SR) compared to directly feeding VLM features into the action expert.

---

### Author Response · Authors · 2025-12-03
**A Summary of Our Rebuttal and Reviewers' Corresponding Comments**

### Reviewer MEHz
**Concerns on:** (1) Limited novelty of spatial priors; (2) Dataset curation feels similar to prior detector-based pipelines; (3) Dual-system design is not new.

**Our responses and clarifications include:** (1) Our contribution is showing how to scale them effectively in end-to-end VLA training so that they produce consistent and significant gains；(2) Our dataset curation includes an unique simulation pipeline built from large-scale 3D assets (14K+ objects), 200+ tabletop configurations, and 1,676 textures; (3) We do not position the dual-system as a contribution.


### Reviewer uJaW
**Concerns on:** (1) Spatial priors are not novel；(2) Lack comparison with spatial-prior VLAs; (3) Real-World setting: DROID or not, and why π₀’s poor performance seems inconsistent with prior experience.

**Our responses and clarifications include:** (1) Our contribution is a scalable, end-to-end spatially guided training recipe, and explained how SP-VLA technically differs from previous spatial VLAs；(2) Added direct comparisons with spatial-aware VLAs on shared benchmarks SimplerEnv and LIBERO；(3) Clarified the real-world setup (non-DROID Franka, detailed in Appendix E.3/E.5) and explained π₀’s lower performance as a consequence of our cluttered setting (10–20 distractors + complex language) and π₀’s lack of large-scale multimodal co-training.


### Reviewer QqBv
**Concerns on:** (1) Source of performance gains; (2) Details on symbolic outputs and spatial prompts; (3) Fairness of data usage in SimplerEnv; (4) Typos and table abbreviations; (5) More details on real-world experiments.

**Our responses and clarifications include:** (1) Three-fold evidence isolating the method's contribution: experiments with a weaker VLM backbone, analysis on baselines using stronger backbones, and controlled ablations with identical backbones; (2) Mathematical formulations and loss of stage 1 and 2, with a detailed description of lightweight latent spatial prompting; (3) Confirmation of identical training data protocols; (4) Clarifications on typos and abbreviations; (5) Comprehensive setup details: the exact instruction-rephrasing prompt, definitions of unseen object position and oriratation, unseen instruction, and explanation of long-horizon tasks autonomy.


### Reviewer BHsq
**Concerns on:** (1) Convergence analysis in SimplerEnv; (2) Analysis of spatial prior data scaling; (3) Missing related work citations; (4) Clarification on the evaluation method of Figure 3(a); (5) Co-training and supervision in stage 2; (6) Plan for releasing the custom simulation benchmark.

**Our responses and clarifications include:** (1) Extended all baselines to 100k steps on WidowX and Google Robot, showing baselines saturate while SP-VLA reaches a higher ceiling; (2) Added scaling results revealing non-linear gains with sharp improvements beyond 2.0M spatial pairs; (3) Added discussions and citations for the requested related works to the related works section; (4) Confirmed that Box IoU@0.5 is measured on the validation set using spatial-grounding prompts during Stage 2 post-training; (5) Clarified that our method inherently integrates co-training, utilizing multimodal data in both stages. Corrected the misunderstanding about Stage 2 supervision. (6) Confirmed the open-sourcing of the custom simulation benchmark and code upon acceptance.

### Reviewer qHTz
**Concerns on:** (1) Missing citations for GR00T, $\pi 0$ and undefined DiT; (2) Insufficient training and architecture details for reproducibility; (3) Detailed loss weighting and RefCOCO-g evaluation; (4) Component ablation; (5) Spatial prompt formats and prompt ablations; (6) Typographical and formatting issues;  (7) Justification for stage 1 grounding pre-training; (8) Unexpectedly strong Vanilla VLA baseline.

**Our responses and clarifications include:**  (1) Added missing citations and expanded acronyms (DiT); (2) Added two-stage loss definitions and architectural comparison details; (3) Reported loss weights and learning rates; clarified RefCOCO-g protocol; (4) Added controlled ablations on planner and actor components; (5) Clarified unified prompt usage and added prompt-format ablations showing SOTA performance; (6) Fixed typos and improved tables and figures; (7) Explained stage 1 as specialized grounding pre-training for high-precision robotics; (8) Attributed stronger Vanilla VLA to the Qwen2.5-VL backbone with the same data protocol.

**Reviewer's corresponding comments:** "I had many technical questions, and most have been adequately addressed. I appreciate the additional ablations, clarifications, typo fixes, and improved presentation; I think these additions help strengthen the paper’s contributions."

---

### Author Response · Authors · 2025-12-03
**General Response**

We sincerely thank all reviewers for their insightful and constructive feedback. We appreciate the positive assessment that **SP-VLA is a sound and effective framework** for scaling spatial priors in end-to-end robot learning. We also thank all reviewers for recognizing that **our experiments are highly solid**, and we especially thank reviewer MEHz for highlighting that **SP-VLA's practical insights could be valuable** to the research community.

Across the reviews, the main concerns can be summarized into two points, and we provide our responses accordingly:

`1. Novelty concern on spatial priors. [MEHz, uJaW]`

We agree that spatial priors are not new concepts. However, our contribution is not the use of spatial priors, but showing how to scale them effectively in end-to-end VLA training so that they produce consistent and significant gains.

In the response to [MEHz, uJaW], we directly clarify SP-VLA has key differences from each related work mentioned by the reviewers.
While prior related works certain spatial signals, SP-VLA is the first to demonstrate consistent improvements under large-scale instruction-following evaluations, especially across both (i) large simulation settings (3,000+ objects) and (ii) real-world conditions in clustered scenes (10–20 distractors + complex language) and long-horizon tasks (more than 10 subtasks).

Analogous to how the GPT series demonstrated that scaling strategies, which do not involve architectural novelty, our work identifies a practical, scalable recipe for integrating spatial priors into modern end-to-end VLA systems.


`2. Concerns about performance gains are mainly attributed to strong backbone[QqBv, qHTz]`


We included a clear step-by-step ablation study in the original paper (Sec. 3.2), and the improvements are evident (54.7→73.2 on WidowX) even over strong baselines using the same VLMs.

To further address backbone dependence, we rebuilt SP-VLA using Florence-2, which is a weaker VLM compared to Qwen2.5-VL and GR00T-VLM. In this setting, SP-VLA still improves the average performance from 46.1 (Vanilla VLA) to 67.9, which exceeds the reported performance of GR00T N1.5.

---

### Meta-Review · Area_Chair_KAN6 · 2026-01-09

**Summary:**

In my opinion, the biggest concerns raised by the reviewers were:

- Lack of novelty in the concept of spatial priors in the context of VLA training.

- Lack of comparison to other VLAs that explicitly try to use spatial priors.

- Insufficient level of detail about the architecture and the experiments.

- Insufficient ablations to attribute the performance gains specifically to the proposed process of incorporating spatial priors.

The rebuttals responded to each of these, as well as a number of smaller concerns, in a lot of detail, including by providing a lot of extra experimental results. While only two of the reviewers responded, their responses indicated that they considered their concerns mostly addressed. Having read all the reviews, all the rebuttals, and the submission itself as an AC, I believe this holds for other reviews too - the authors' responses addressed them all well by clarifying the contributions and providing valuable extra results.

One aspect that could have potentially given some reviewers pause even if they had a chance to respond to the rebuttals is that, as the authors acknowledge, spatial priors aren't the paper's innovation. In my mind, however, this is fine. The paper's main contribution is a recipe, supported by thorough empirical evidence, for consistently getting mileage out of spatial priors when training VLAs. Given the solid experiments (especially with the improvements the authors have made for the rebuttal) and thematic alignment with ICLR's main topics, this contribution is definitely valuable enough to merit acceptance at ICLR.

**Reviewer Concerns:**

Please see above.

**Reviewer Scores:**

Out of reviewers MEHz, qHTz, and BHsq (all originally gave a 6, and qHTz, and BHsq also responded positively to the rebuttals) I believe at least one and possibly two would have increased their scores to 7

Reviewers uJaW and QqBv (both originally gave a 4) would likely have increased their scores as well, but probably to 5 or 6.

---

### Decision · Program_Chairs · 2026-01-26

Accept (Poster)